# Uncertainty-Based Smooth Policy Regularisation for Reinforcement Learning with Few Demonstrations

**Yujie Zhu[1], Charles A. Hepburn[1], Matthew Thorpe[1], and Giovanni Montana[1,2]**

[1]Department of Statistics, [2]Warwick Manufacturing Group
University of Warwick
CV4 7AL
{yujie.zhu, charles.hepburn, matthew.thorpe, g.montana}@warwick.ac.uk

## Abstract

In reinforcement learning with sparse rewards, demonstrations can accelerate learning, but determining when to imitate them remains challenging. We propose Smooth Policy Regularisation from Demonstrations (SPReD), a framework that addresses the fundamental question: *when should an agent imitate a demonstration versus follow its own policy?* SPReD uses ensemble methods to explicitly model Q-value distributions for both demonstration and policy actions, quantifying uncertainty for comparisons. We develop two complementary uncertainty-aware methods: a probabilistic approach estimating the likelihood of demonstration superiority, and an advantage-based approach scaling imitation by statistical significance. Unlike prevailing methods (e.g. Q-filter) that make binary imitation decisions, SPReD applies continuous, uncertainty-proportional regularisation weights, reducing gradient variance during training. Despite its computational simplicity, SPReD achieves remarkable gains in experiments across eight robotics tasks, outperforming existing approaches by up to a factor of 14 in complex tasks while maintaining robustness to demonstration quality and quantity. Our code is available at https://github.com/YujieZhu7/SPReD.

## 1 Introduction

Reinforcement learning (RL) has proven effective for sequential decision problems in robotics [1, 2] and games [3, 4], yet complex tasks require extensive iterations that introduce risks and costs when learning must occur in real-world settings. To accelerate learning, researchers have incorporated pre-collected demonstrations [5, 6, 7], particularly valuable for sparse-reward environments where agents struggle with minimal feedback. While dense reward functions could help, they require domain expertise and become increasingly difficult to design for complex dynamics. Sparse rewards, though simpler and less susceptible to local optima [5], significantly intensify the exploration challenge—making demonstrations essential for effective learning.

The field has produced numerous approaches to leverage demonstrations in RL, with varying degrees of success. Early methods employ prioritised replay mechanisms but require extensive parameter tuning and struggle with demonstration quality adaptation [5, 8]. Other techniques use weighted behaviour cloning (BC) with predetermined decay factors that reduce demonstration influence over time regardless of their continuing utility [7]. More recent methods attempt to address limited or suboptimal demonstrations through reduced Q-values for undemonstrated actions [9] or variance estimates as weighted guidance signals [10], but are restricted to discrete action spaces. Jing et al. [11] treat demonstrations as a soft constraint on policy exploration, formulating a constrained policy optimisation problem. Although they reduce overhead by applying a local linear search on its

39th Conference on Neural Information Processing Systems (NeurIPS 2025).

dual, the approach still involves considerable computational complexity. Efficiently leveraging such demonstrations remains an underexplored problem.

The Q-filter approach [6] brought significant advancements by selectively applying behaviour cloning only when demonstration actions yield higher Q-values than policy actions. This intuitive method has become foundational in RL with demonstrations, yet our analysis reveals two critical limitations: it relies on single point estimates without accounting for estimation uncertainty, and it makes binary decisions that introduce high variance during training. These limitations become particularly problematic with limited or suboptimal demonstrations—common scenarios in practical applications.

To directly address these fundamental limitations of Q-filter, we propose Smooth Policy Regularisation from Demonstrations (SPReD), which reformulates demonstration utilisation as a distributional comparison problem. Our approach employs ensembles of critics to model two distinct Q-value distributions: one evaluating demonstration actions and another evaluating current policy actions. Unlike Q-filter's binary approach, SPReD applies continuous weights to the behavioural cloning loss, determining how strongly each demonstration action should influence policy updates. These weights scale smoothly with our statistical confidence that demonstration actions outperform the current policy. We develop two complementary methods for this value distribution comparison: a probabilistic weighting based on the likelihood that demonstration actions are superior, and an exponential scaling inspired by weighted behavioural cloning that calibrates imitation strength based on the statistical significance of advantages. Both methods yield state-adaptive regularisation that enables smoother policy learning across diverse environments.

Our theoretical analysis establishes several key properties: continuous weights strictly reduce policy gradient variance compared to binary decisions; they adapt systematically to uncertainty levels; and they progressively diminish influence from suboptimal demonstrations as policy performance improves. These properties provide a sound foundation for the empirical improvements we observe. Through experiments across eight robotic tasks, we demonstrate that SPReD consistently outperforms existing methods, achieving up to 14× success rates with the same interaction steps in complex manipulation tasks like block stacking (0.920 vs. 0.064) and significant improvements even with severely limited or suboptimal demonstrations. Despite using ensembles, our implementation maintains efficiency comparable to standard methods by leveraging the same critic networks for both target computation and uncertainty estimation. Our results demonstrate that our uncertainty-aware approach to comparing value distributions enables effective learning from limited or noisy demonstrations.

## 2   Related work

**RL from expert demonstrations.**   Expert demonstrations enhance online RL across various approaches [12]. Atkeson and Schaal [13] pioneered demonstration-based task model and reward function learning as initialisation for policy learning. More advanced methods leverage demonstrations throughout the entire learning process. DDPGfD [5] and DQfD [8] utilise prioritised replay buffers combining demonstrations and interactions, but require extensive parameter tuning and lack mechanisms for handling suboptimal demonstrations. DAPG [7] implements an on-policy method with weighted BC loss based on the advantage function, but its influence diminishes through a non-adaptive decay factor regardless of demonstration quality. Our method differs by employing weighted BC with adaptive Q-value comparisons specific to each state-action pair, enabling smooth regularisation that effectively incorporates both expert and suboptimal demonstrations—addressing a key limitation of previous approaches.

**RL from suboptimal demonstrations.**   Recent research focuses on handling imperfect demonstrations adaptively. Nair et al. [6] introduced the Q-filter by incorporating demonstrations in DDPG without pretraining, using BC loss filtered by estimated Q-values to selectively imitate demonstrators. Our ensemble method directly enhances this concept through uncertainty quantification. Other approaches include Assisted DDPG [14], which relies on external controllers, LIDAR [15], which considers the advantage of demonstrations, and NAC [9], which reduces Q-values of unobserved demonstration actions in discrete spaces. While some methods require demonstration pretraining [9], our approach remains sample efficient without this step. Alternative approaches include constrained policy optimisation [11], BQfD's variance-based guidance [10], and RLPD [16], which uses symmetric buffer sampling and ensemble critics with Layer normalisation. We select Q-filter and RLPD as

primary baselines given their comparable settings—both handle suboptimal demonstrations without pretraining and provide state-of-the-art performance.

**Offline-to-online RL.** Offline RL enables learning policies that surpass static dataset performance without online interactions [17, 18, 19], but remains constrained by dataset coverage and quality. Offline-to-online RL addresses this through subsequent fine-tuning, facing challenges with inaccurate Q-value estimates for out-of-distribution state-action pairs [20]. IQL [21] uses expectile regression to learn a value function and performs advantage-weighted policy extraction to learn a policy without explicit bootstrapping from the policy. Methods like AWAC [22] implement fine-tuning through implicit policy constraints, while others gradually relax BC constraints [23], employ ensemble pretraining with pessimistic Q-functions [20], or generate interactions from composite policies [24]. Unlike our approach, these methods typically require extensive offline datasets for pretraining, whereas our method operates effectively with few demonstrations in a purely online manner.

**Uncertainty in RL** Uncertainty in RL is typically categorised as aleatoric (inherent environmental randomness) or epistemic (model knowledge limitations) [25]. While uncertainty quantification has proven valuable for exploration-exploitation balancing [26, 27], safety constraints [28], and offline learning [29], its application to demonstration utilisation remains underdeveloped. Common approaches for uncertainty estimation include bootstrapping [30], ensemble techniques [31], and MC-dropout [32], with some methods explicitly addressing both uncertainty types [33]. UWAC [34] is an offline RL method that weights critic and actor updates using dropout-based uncertainty to manage out-of-distribution actions. Most online RL approaches employing uncertainty with demonstrations—such as Active DQN [35], RCMP [36], and CHAT [37]—make binary decisions about demonstration usage. Our work differs by using ensemble-based epistemic uncertainty estimates to enable smooth, continuous regularisation based on quantified confidence in demonstration superiority.

## 3  Preliminaries

**Setup** We consider the standard Markov decision process (MDP) framework $\mathcal{M} = (\mathcal{S}, \mathcal{A}, P, R, \gamma, \rho_0)$ where an agent interacts with an environment $E$ over discrete time steps. The initial state distribution is $\rho_0$. At each time step, the agent observes state $s \in \mathcal{S}$, selects action $a \in \mathcal{A}$ according to a policy $\pi$, and receives reward $r_t = R(s, a)$ while transitioning to the next state $s'$ according to the environment dynamics $P(s'|s, a)$. The transition tuples $(s, a, r, s')$ are saved in replay buffer $\mathcal{B}$ with exploration noise added to actions, and mini-batches are sampled from it for future learning. With discount factor $\gamma \in [0, 1)$, the agent aims to maximise the expected cumulative discounted return $J = \mathbb{E}_{r_i, s_i \sim E, a_i \sim \pi}[R_0]$ where $R_t = \sum_{i=t}^{T} \gamma^{i-t} r_i$. The state-action value function is defined as $q_\pi(s, a) = \mathbb{E}_\pi[R_t|s, a]$, with the estimate $Q^\pi(s, a)$. Additionally, we assume access to a set of demonstrations $\mathcal{D} = \{(s_d, a_d, r_d, s'_d)\}$ collected from an unknown policy $\pi_D$, which may be suboptimal. Our goal is to effectively leverage these demonstrations to accelerate learning.

**TD3 and HER** Our method is compatible with any off-policy actor-critic RL algorithm. For our experiments, we implement SPReD with the state-of-the-art TD3 [38] algorithm, which provides an ideal foundation due to its model-free nature and suitability for continuous state and action spaces. Critics are updated by minimising the mean squared error:

$$\mathcal{L}(\theta_i) = \mathbb{E}_{(s,a)\sim\mathcal{B}}(r + \gamma \min_{i=1,2} Q_{\theta'_i}(s', \tilde{a}) - Q_{\theta_i}(s, a))^2,$$

where dual critic networks with target networks are employed in the target to address the overestimation bias [39, 38], and $\tilde{a}$ is the action selected by the target actor with Gaussian noise. The actor parameters $\phi$ are updated using deterministic policy gradient [40] to maximise:

$$J(\phi) = \mathbb{E}_{(s,a)\sim\mathcal{B}} Q_{\theta_1}(s, \pi_\phi(s)).$$

For environments with sparse rewards, we employ Hindsight Experience Replay (HER) [41] to address exploration challenges in goal-conditioned tasks. HER stores transitions twice with desired goals or actually achieved goals, enabling learning from unsuccessful episodes. The goals are appended to states as inputs for actor and critic networks.

**Q-filter** To leverage demonstrations for accelerating learning, the Q-filter [6] stores demonstration data in a separate buffer $\mathcal{B}_D$. During each training step, a mini-batch of size $N_D$ is sampled from the demonstration buffer in addition to transitions from the standard replay buffer. Both are used for critic updates. The Q-filter technique incorporates a selective BC loss that only imitates demonstration actions when they are estimated to be superior to the current policy's actions:

$$L_{BC}(\phi) = \mathbb{E}_{(s_d, a_d) \sim \mathcal{B}_D}[\|\pi_\phi(s_d) - a_d\|^2 \mathbb{1}_{Q(s_d, a_d) > Q(s_d, \pi_\phi(s_d))}],$$

where $\mathbb{1}$ is the indicator function that equals 1 when the single Q estimate of the demonstration action is higher and 0 otherwise. The actor network is then updated by optimising the combined objective, $-\lambda_1 J + \lambda_2 L_{BC}$, where $\lambda_1$ and $\lambda_2$ are hyperparameters that balance policy improvement against demonstration imitation.

## 4 Methodology

The Q-filter mechanism [6] represents a state-of-the-art approach for demonstration utilisation in reinforcement learning, particularly for continuous control tasks with sparse rewards. While effective, this approach makes binary decisions to accept or reject demonstrations based on point estimates of Q-values. We hypothesised that such binary filtering mechanisms fundamentally limit learning efficiency in two ways: they fail to account for estimation uncertainty inherent in temporal difference learning, and they introduce gradient discontinuities during policy updates. These limitations would theoretically cause increasingly unstable learning as policies approach optimality, precisely when nuanced guidance from demonstrations becomes most valuable.

To address these limitations, we reformulate demonstration-based regularisation by directly comparing the quality of demonstration actions versus current policy actions while accounting for uncertainty. For each state-action pair in the demonstration buffer, we compare two distributions: the distribution of Q-values for the demonstration action $\{Q_i(s_d, a_d)\}_{i=1}^m$ and the distribution of Q-values for the current policy action $\{Q_i(s_d, \pi_\phi(s_d))\}_{i=1}^m$, where $i \in [1, m]$ indexes our ensemble of $m$ independent critic networks. The variability across these ensemble estimates captures the epistemic uncertainty in our value approximation. The selection of uncertainty measure is discussed in Appendix F.

**Smooth policy regularisation from demonstrations** Rather than using point estimates as in Q-filter, we leverage the full distributions of Q-values from our ensemble to determine how strongly each demonstration should influence policy learning. The key insight of our approach is that demonstration influence should vary continuously with our confidence in its superiority. Rather than making binary decisions, we introduce a state-adaptive weight $p \in [0, 1]$ that quantifies our statistical confidence that a demonstration action outperforms the current policy action. This weight modulates a behaviour cloning loss:

$$L_{\text{WBC}} = \mathbb{E}_{(s_d, a_d) \sim \mathcal{B}_D}[p(s_d, a_d)\|\pi_\phi(s_d) - a_d\|^2],$$

where higher $p$ values (the dependency on $(s_d, a_d)$ is dropped in notation for simplicity) apply stronger regularisation toward demonstration actions when we are more confident in their superiority, and lower values reduce imitation pressure when demonstrations appear less valuable.

Following the TD3 framework [38], we then update the actor network by combining standard deterministic policy gradient with this weighted behaviour cloning term:

$$\mathcal{L}(\phi) = -\lambda_1 \mathbb{E}_{(s,a) \sim \mathcal{B}} \bar{Q}_\theta(s, \pi_\phi(s)) + \lambda_2 L_{\text{WBC}}. \tag{1}$$

This continuous weighting mechanism creates a smooth regularisation effect that adapts to the varying quality of demonstrations while accounting for uncertainty in value estimates. Despite its computational simplicity, this approach offers theoretical benefits (see Section 5) and remarkable empirical performance (see Section 6).

The central challenge now becomes: *how should we compute this weight $p$ by comparing two value distributions?* We present two complementary approaches for this distributional comparison: a probabilistic method that estimates the likelihood of demonstration superiority, and an exponential method that scales imitation strength based on the statistical significance of advantages. Despite their different formulations, these methods are theoretically connected, with similar behaviour in high-uncertainty regimes as we demonstrate in Property 5.3.

**SPReD-P: Probabilistic advantage weighting**    Our first method, SPReD-P, frames the above comparison as a probabilistic inference problem. Following common practice in uncertainty quantification, we model the Q-value estimates from our ensemble as Gaussian distributions [27, 42, 43]:

$$Q(s_d, a_d) \sim \mathcal{N}(\hat{Q}(s_d, a_d), \hat{\sigma}_d^2)$$

$$Q(s_d, \pi_\phi(s_d)) \sim \mathcal{N}(\hat{Q}(s_d, \pi_\phi(s_d)), \hat{\sigma}^2)$$

where $\hat{Q}(s_d, a_d)$, $\hat{Q}(s_d, \pi_\phi(s_d))$ and $\hat{\sigma}_d^2$, $\hat{\sigma}^2$ represent the empirical mean and variance of Q-value estimates across the ensemble. With these distributions established, we compute the probability that the demonstration action outperforms the current policy action:

$$p_P = \mathbb{P}(Q(s_d, a_d) > Q(s_d, \pi_\phi(s_d))) = \Phi\left(\frac{\hat{Q}(s_d, a_d) - \hat{Q}(s_d, \pi_\phi(s_d))}{\sqrt{\hat{\sigma}_d^2 + \hat{\sigma}^2}}\right)$$

where $\Phi$ represents the cumulative distribution function of the standard normal distribution.

This formulation creates a continuous spectrum of imitation strengths that naturally adapts to uncertainty levels throughout training. Early in training when Q-value estimates have high variance, probabilities tend toward intermediate values ($\approx 0.5$), allowing partial learning even from uncertain examples. As uncertainty decreases with more training, the probabilities become more decisive, approaching the binary case only when uncertainty becomes negligible; see Property 5.1 and empirical evidence in Appendix F. The Gaussian assumption provides a computationally efficient approximation while remaining analytically tractable; empirical comparisons in Appendix F confirm this assumption is plausible in practice.

**SPReD-E: Exponential advantage weighting**    While SPReD-P provides a principled probabilistic approach, we now introduce SPReD-E, which focuses on the *magnitude of improvement* rather than its likelihood. This complementary method scales imitation strength based on how significantly a demonstration outperforms the current policy relative to estimation uncertainty. The core insight is that imitation should be proportional to the size of the advantage, accounting for Q-value variability.

As before, using our ensemble of critics, we generate two distributions of Q-values for each state in the demonstration buffer: the distribution of demonstration Q-values and the distribution of current policy Q-values. To quantify how much better the demonstration is compared to the current policy, we need to derive an advantage measure $A$ that respects the distributional nature of our estimates.

To simplify notation, let $\mu$ and $\nu$ represent the Q-value distributions under the current policy and demonstration, respectively. Heuristically, we aim to define a weight $p_E$ that follows the current policy when its Q-values exceed those of demonstrations, and increasingly imitates demonstrations as their relative advantage grows. If the Q-value was a single value (that is, a Dirac distribution), then we could define $p_E$ as a function of the difference, say $A$, between the values, for example $p_E = e^{A/\beta} - 1$. To treat measures we look at comparing $x$ in the support of $\mu$ to $y$ in the support of $\nu$. Formally, this is done through a transport map $T$ that rearanges $\mu$ to form $\nu$ (which can be written $\nu = T_\# \mu := \mu(T^{-1}(\cdot))$). We define $A = \int (T(x) - x) \, \mathrm{d}\mu(x)$ to be the average difference between Q-values given the map $T$. Since (by a change of variables) we can write $A = \mathbb{E}_\mu[Q] - \mathbb{E}_\nu[Q]$, there is no dependence on $T$ and $A$ is simply the difference between means. This short argument justifies computing the advantage directly as the difference of mean Q-values:

$$A = \mathbb{E}_{i\in[m]}\left[Q_i(s_d, a_d)\right] - \mathbb{E}_{i\in[m]}\left[Q_i\left(s_d, \pi(s_d)\right)\right].$$

Having derived this advantage measure, we transform it into a weight using an exponential function inspired by advantage-weighted behavioural cloning [44] and policy improvement techniques:

$$p_E = e^{A/\beta} - 1$$

where $\beta$ controls sensitivity to advantage magnitude. We use a proportion of the interquartile range (IQR) of Q-value distributions as $\beta$ to capture uncertainty in advantage estimates, providing robustness against outliers while achieving state-adaptive normalisation. This formulation is clipped to the range $[0, 1]$ to ensures zero weight for inferior demonstrations ($A < 0$), proportional weighting for uncertain cases (small $|A|$), and strong imitation for clearly superior demonstrations (large $A$). The subtraction of 1 creates a natural zero-threshold exactly when demonstration and policy actions are equally valuable.

# 5 Theoretical properties

We now establish the theoretical foundations of SPReD, examining how both weighting mechanisms adapt to varying uncertainty conditions and demonstrating their advantages over binary filtering; full proofs are provided in Appendix B and empirical evidence are provided in Appendix F. Our analysis focuses on key properties that characterise the behaviour and benefits of continuous weighting.

First, we formalise the fundamental advantage of continuous weights over binary decisions:

**Lemma 5.1** (Gradient-variance gap). *Assuming (A1) gradient norms are bounded and (A2) demonstrations are independently sampled, let $X_k = \mathbb{1}_k g_k$, $Y_k = p_k g_k$, $g_k = \nabla_\phi \|\pi_\phi(s_k) - a_k\|^2$. Then*

$$\mathrm{Var}\Big[\frac{1}{N_D}\sum_k Y_k\Big] \leq \mathrm{Var}\Big[\frac{1}{N_D}\sum_k X_k\Big],$$

*where $\mathbb{1}_k$ represents binary filtering decisions, $p_k \in [0,1]$ represents our continuous weights, and $N_D$ is the batch size. Strict inequality holds if $\mathbb{P}(0 < p_k < 1) > 0$.*

This result establishes that SPReD's smooth weighting produces BC gradient estimates with lower variance than binary Q-filter approaches. This variance reduction directly improves training stability and sample efficiency [45, 46, 47, 48], particularly in the early stages of learning when policy updates are most sensitive to demonstration influence.

Next, we characterise how our weights respond to different uncertainty conditions:

**Property 5.1** (Adaptive behaviour). *Assume $\beta = \alpha\hat{\beta}$ where $\hat{\beta}$ is the IQR of the mixture model with components $Q(s_d, a_d)$ and $Q(s_d, \pi_\phi(s_d))$. Let $\hat{\sigma}_d^2$ and $\hat{\sigma}^2$ be the variances and $A$ be the difference of means under assumptions given in Appendix B. As this variance varies, our weights satisfy:*

  (i)  High-certainty: *If $\hat{\sigma}^2 + \hat{\sigma}_d^2 \to 0$ then $p_P \to \mathbb{1}_{A>0}$ (with $p_P = 0.5$ if $A = 0$) and $p_E \to \mathrm{clip}(e^{\frac{1}{\alpha}} - 1, 0, 1)$ if $A > 0$ (with $p_E = 0$ if $A \leq 0$).*

  (ii)  High-uncertainty: *If $\hat{\sigma}^2 + \hat{\sigma}_d^2 \to \infty$ then $p_P \to 0.5$ and $p_E \to 0$.*

This property demonstrates how both methods adapt to uncertainty: approaching binary filtering when confident (accepting only positive advantages, with $p_E = 0$ for negative advantages), while becoming conservative under high uncertainty. This adaptive behaviour is crucial for robust learning throughout training.

The following property addresses how SPReD handles potentially suboptimal demonstrations as learning progresses:

**Property 5.2** (Diminishing weight on suboptimal demonstrations). *Let $Q_t$ be the Q-value distribution corresponding to the policy $\pi_t$. Assume $\pi^*$ is the optimal policy with Q-value function $Q^*$. We assume that there is no uncertainty in $Q^*$ so that $Q^*$ is a single value function (not a distribution). Let $(s_d, a_d)$ satisfy $Q^*(s_d, a_d) < Q^*(s_d, \pi^*(s_d))$. Assume (A3) the mean $\hat{Q}_t(s, a)$ of the random variable $Q_t(s, a)$ converges to $Q^*(s, a)$ and the variance $[\hat{\sigma}_d]_t^2 \to 0$ uniformly over actions (so that the policy $\pi_t$ is in a sense converging to the optimal policy $\pi^*$). Then*

$$\lim_{t\to\infty} p_P(s_d, a_d) = 0, \quad \lim_{t\to\infty} p_E(s_d, a_d) = 0.$$

Crucially, this property enables SPReD to autonomously down-weight inferior demonstration actions as the policy improves and uncertainty decreases, allowing the agent to filter out misleading information and potentially surpass the performance of provided demonstrations without requiring explicit scheduling or quality estimation.

Finally, we establish a fundamental connection between our two weighting schemes:

**Property 5.3** (Parameter scaling relationship). *When advantage magnitudes are small compared to estimation uncertainty ($|A|/\sigma \ll 1$ where $\sigma = \sqrt{\hat{\sigma}_d^2 + \hat{\sigma}^2}$), Taylor expansion yields:*

$$p_P = \frac{1}{2} + \frac{A}{\sigma\sqrt{2\pi}} + \mathcal{O}\big((A/\sigma)^3\big), \quad p_E = \frac{A}{\beta} + \mathcal{O}\big((A/\beta)^2\big).$$

*This reveals that setting $\beta = \sigma\sqrt{2\pi} \approx 2.5\sigma$ causes both methods to exhibit similar rates of change with A in high-uncertainty scenarios, precisely when smooth regularisation is most beneficial. Under*

*this parameterisation, both approaches implement proportional forms of uncertainty-aware caution up to a constant, differing only in higher-order terms.*

This relationship reveals that our exponential weighting provides a non-parametric alternative that achieves similar uncertainty-aware adaptation without requiring Gaussian assumptions, while naturally placing greater emphasis on demonstrations with larger advantages. The theoretical connection also provides principled guidance for parameter selection (see Appendix B for more details).

## 6 Experimental results

**Environments and tasks**   We evaluate SPReD on eight challenging robotics tasks from OpenAI Gym's Fetch and Shadow Dexterous Hand environments [49], simulated in MuJoCo [50]. These environments feature sparse binary rewards (feedback only upon goal completion) and complex multi-goal structures, creating significant exploration challenges that make them particularly suitable for demonstration-based learning. The Fetch tasks utilise a 7-DoF robotic arm with parallel gripper for pushing, sliding, pick-and-place, and block stacking operations of increasing difficulty. For the stacking tasks, we use implementations and expert policies from Lanier [51]. The Shadow Hand tasks represent substantially higher complexity, requiring a 24-DoF anthropomorphic hand to achieve precise rotational control of objects (block, egg, pen) despite high-dimensional action spaces and control noise sensitivity. We exclude the trivial FetchReach task and use 1000 demonstration episodes for the challenging 3-block stacking task, with 100 demonstrations for all other tasks. See Appendix D and Appendix E for additional details about environments and demonstrations. We also experiment on locomotion tasks [52] with dense rewards for more evaluation domains in Appendix F.

**Baselines**   We evaluate SPReD against several state-of-the-art baselines, all implemented with HER for fair comparisons. Our primary RL baseline is **TD3** [38], which operates without demonstration utilisation. We include **EnsTD3**, an ensemble version using 10 critics where random pairs compute minimum target values and the ensemble mean guides actor updates (similar to REDQ [53]), isolating ensemble effects without demonstrations. We compare against **Q-filter**, the approach from Section 3 using binary-filtered BC with point Q-value estimates [6], and its ensemble variant **EnsQ-filter** that uses critic means for BC decisions, helping isolate benefits beyond simple ensembling. We evaluate against **RLPD** [16], a recent method leveraging ensemble critics with layer normalisation, implemented with author-recommended hyperparameters and input normalisation for optimal performance. We also include three variants of AWAC, a state-of-the-art offline-to-online RL method based on advantage weights: **AWAC** with no pretraining on prior data, **AWAC-p** with pretraining, and **AWAC-r** without pretraining but keeping resampling demonstrations. Our proposed SPReD approach is implemented in the two variants: **SPReD-P** and **SPReD-E**. Both methods leverage ensemble critics to quantify uncertainty but differ in how they transform uncertainty estimates into imitation weights. The pseudocode of our method and training details with computational cost analysis can be found in Appendix C. The ablation tests on ensemble size, isolated contribution of different components, and normalisation constant of SPReD-E are presented in Appendix G.

**Main results**   Table 1 presents success rates after 1 million interactions (10 million for challenging stacking tasks), quantifying sample efficiency across methods. Our uncertainty-aware methods (SPReD-P and SPReD-E) consistently outperform standard Q-filter, its ensemble variant, RLPD and all variants of AWAC across all tasks, with SPReD-E achieving significantly higher success rates in seven of eight environments than baselines, and both methods are stable exhibiting relatively lower variance. AWAC struggles significantly in our setting with few demonstrations and environments with sparse rewards as SPReD explicitly reasons about when demonstrations remain useful rather than relying on advantage estimates from limited data.

The sample efficiency advantage of our approach increases with task complexity and is most pronounced in intricate manipulation tasks. In FetchStack2, our methods achieve 14× the success rate of RLPD (0.920 vs. 0.064), despite using only 100 demonstrations. Even in the challenging sliding task, where demonstrations provide limited benefit due to sensitivity to precise force application, SPReD-E achieves a 50% higher success rate (0.240) than the next best method (0.160).

The entire learning curves in Figure 1 provide more comprehensive performance comparisons extending beyond the 1-10 million interactions reported in Table 1, showing longer-term behavior. Apart from the substantial improvement of sample efficiency, our methods match or exceed all baselines

Table 1: Average success rate (with standard deviation) over 5 seeds after 1M environment interactions (10M for stacking tasks). The highlighted results lie between the mean of the best performer and one standard deviation below it (i.e., if the best result is $\mu \pm \sigma$, all values $\geq \mu - \sigma$ are bold). The success rates of demonstrations range from 0.2-0.86 for standard tasks and 1.0 for stacking tasks.

| Environment | Methods | | | | | | | | | |
|---|---|---|---|---|---|---|---|---|---|---|
| | TD3 | EnsTD3 | Q-filter | EnsQ-filter | RLPD | AWAC | AWAC-p | AWAC-r | SPReD-P | SPReD-E |
| FetchPush | 0.304 ± 0.272 | 0.272 ± 0.227 | 0.792 ± 0.016 | 0.880 ± 0.044 | **0.968 ± 0.016** | 0.112 ± 0.078 | 0.064 ± 0.032 | 0.280 ± 0.076 | **0.976 ± 0.020** | **0.984 ± 0.032** |
| FetchSlide | 0.040 ± 0.062 | 0.064 ± 0.109 | 0.072 ± 0.039 | 0.104 ± 0.032 | 0.160 ± 0.057 | 0.112 ± 0.117 | 0.032 ± 0.047 | 0.072 ± 0.073 | 0.112 ± 0.096 | **0.240 ± 0.044** |
| FetchPickAndPlace | 0.080 ± 0.036 | 0.192 ± 0.209 | 0.608 ± 0.047 | 0.688 ± 0.069 | 0.640 ± 0.044 | 0.048 ± 0.030 | 0.256 ± 0.090 | 0.488 ± 0.089 | **0.832 ± 0.111** | **0.888 ± 0.064** |
| FetchStack2 | 0.008 ± 0.016 | 0.000 ± 0.000 | 0.048 ± 0.039 | 0.144 ± 0.103 | 0.064 ± 0.048 | 0.008 ± 0.016 | 0.008 ± 0.016 | 0.008 ± 0.016 | 0.840 ± 0.110 | **0.920 ± 0.057** |
| FetchStack3 | 0.000 ± 0.000 | 0.000 ± 0.000 | 0.048 ± 0.059 | 0.040 ± 0.025 | 0.080 ± 0.062 | 0.000 ± 0.000 | 0.000 ± 0.000 | 0.000 ± 0.000 | 0.248 ± 0.120 | **0.384 ± 0.125** |
| ManipulateBlock | 0.072 ± 0.053 | 0.136 ± 0.048 | 0.760 ± 0.025 | **0.856 ± 0.103** | 0.016 ± 0.020 | 0.008 ± 0.016 | 0.008 ± 0.016 | 0.144 ± 0.041 | **0.864 ± 0.065** | **0.832 ± 0.089** |
| ManipulateEgg | 0.000 ± 0.000 | 0.000 ± 0.000 | 0.056 ± 0.054 | 0.096 ± 0.041 | 0.000 ± 0.000 | 0.000 ± 0.000 | 0.000 ± 0.000 | 0.000 ± 0.000 | **0.208 ± 0.082** | 0.16 ± 0.076 |
| ManipulatePen | 0.000 ± 0.000 | 0.080 ± 0.025 | 0.208 ± 0.111 | **0.264 ± 0.093** | 0.000 ± 0.000 | 0.000 ± 0.000 | 0.008 ± 0.016 | 0.000 ± 0.000 | 0.216 ± 0.065 | **0.288 ± 0.053** |

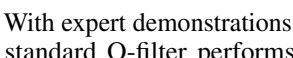

Figure 1: Performance comparison across eight robotics tasks. Solid lines represent mean success rates across 5 seeds, with shaded areas showing standard deviation. The learning curves are smoothed using a 5-point moving average. Horizontal dashed lines indicate the success rates of the demonstrations used for training. Our SPReD methods (red and brown) consistently outperform baselines across environments of varying complexity.

asymptotically. SPReD excels particularly in complex tasks like stacking, where demonstrations are most needed. In some extremely sensitive tasks where considerably suboptimal demonstrations provide limited guidance, EnsQ-filter and RLPD show competitive performance asymptotically but fail to adapt to different tasks.

From a computational perspective, instead of processing 10 critics sequentially, we stack their computations into batched tensor operations that execute in parallel with the vectorised critic. SPReD maintains efficiency and achieves nearly the same throughput as TD3 despite using $5\times$ more critics, while RLPD demands approximately twice the computation time (shown in Appendix C).

**Impact of demonstration quality** To systematically assess robustness to demonstration quality, we conduct experiments at three distinct quality levels in Fetch-PickAndPlace and Fetch-Push as illustrated in Figure 2 and Appendix G.

With expert demonstrations, standard Q-filter performs adequately since most demonstration actions merit imitation. However, its performance drops sharply as demonstration quality decreases, showing the weaknesses of binary filtering when dealing with uncertain Q-value comparisons. In the most extreme case—the PickAndPlace environment with only one expert trajectory among 99 random trajectories (the right plot in Figure 2)—standard Q-filter

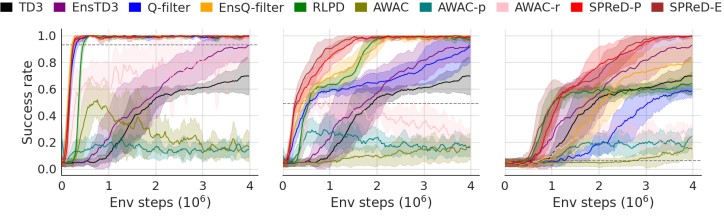

Figure 2: Effect of demonstration quality in FetchPickAndPlace. The demonstrations are expert, suboptimal and severely suboptimal from left to right with success rates shown as dashed lines.

actually performs worse than TD3 without demonstrations, effectively amplifying misleading examples rather than filtering them. RLPD neither fully leverage the advantage of expert demonstrations nor discard the sub-optimality. In contrast, both SPReD variants maintain consistent performance across all quality levels. The automatic, continuous adaptation mechanism shown in Property 5.1 and Appendix F enables SPReD methods to extract maximum value from demonstrations of any quality level, maintaining superior sample efficiency across all test conditions without requiring explicit scheduling or quality estimation procedures.

**Impact of demonstration sample size** We systematically evaluate how varying the sample size of available demonstrations (episodes) affects learning performance across methods in Figure 3.

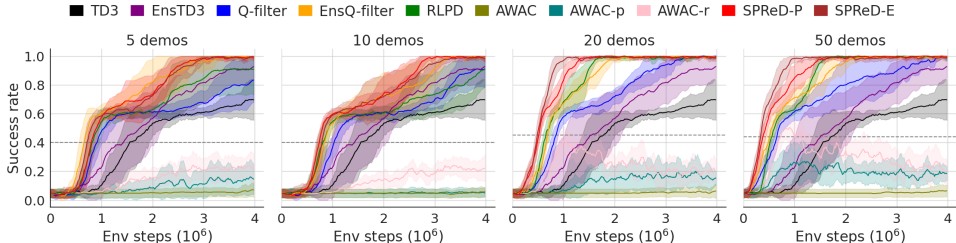

Figure 3: Effect of demonstration size in FetchPickAndPlace. The demonstrations collected from the same policy contain 5, 10, 20, or 50 episodes with success rates shown as the dashed lines.

Our SPReD methods demonstrate superior performance even with extremely limited data (10 demonstrations), while maintaining improvement as sample size increased. With only 5 demonstrations, the ensemble Q-filter approach shows reasonable performance but exhibited substantially higher variance across random seeds, indicating less reliable learning. In general, SPReD-P demonstrates greater robustness to limited sample sizes, while SPReD-E yields better asymptotic performance when provided with either larger sample sizes or higher-quality demonstrations, suggesting a potential trade-off between the two approaches depending on available demonstration characteristics.

**Robustness** To show the robustness of our SPReD methods in comparison to the baseline, we evaluate our methods with noisy rewards, with results shown in Appendix F. SPReD remains the best method against the baselines, solving the task with noisy rewards where other methods fail to learn.

# 7 Conclusion

We have introduced SPReD, a framework for smooth policy regularisation from demonstrations that enhances RL in sparse-reward environments. Our key contribution is a principled approach to uncertainty-aware demonstration utilisation through ensemble-based Q-value modelling. We developed two complementary weighting methods: SPReD-P, which leverages probabilistic estimates of demonstration superiority, and SPReD-E, which scales imitation strength based on the statistical significance of advantages. Both methods significantly reduce policy gradient variance compared to binary filtering approaches. Our extensive evaluation across eight robotics tasks demonstrates substantial performance improvements, with up to 14× success rates in challenging manipulation tasks while maintaining robustness to varying demonstration quality and quantity. Based on our results, we recommend SPReD-E as the default choice for most applications, particularly for complex tasks, while noting that both methods achieve significant gains with minimal computational overhead.

Despite these advances, important limitations remain. While SPReD significantly accelerates learning in complex manipulation tasks, we observed that demonstration influence can sometimes slow later-stage learning in highly dexterous tasks, suggesting the need for automatic balancing between demonstration guidance and exploration. As with many deep RL approaches, practical considerations like hyperparameter sensitivity and sample complexity in extremely high-dimensional tasks remain areas for continued refinement. Future work should address the automatic adaptation of demonstration influence throughout training, potentially through meta-learning approaches. Theoretical analysis of convergence properties would also strengthen the foundation of demonstration-based RL methods. Sim-to-real gap is another research direction as our evaluation based on the simulation. Finally, the simplicity of our approach makes it readily applicable to other off-policy RL algorithms beyond TD3, offering potential improvements across a broader range of tasks and domains.

# 8 Acknowledgments

YZ acknowledges funding from the Department of Statistics, University of Warwick. CH acknowledges support from Innovate UK under the project "Fundamental Science and Outreach for Connected and Automated ATM: HyperSolver" (grant no. 101114820). MT would like to acknowledge the support of the Leverhulme Trust through the Research Project Award "Robust Learning: Uncertainty Quantification, Sensitivity and Stability" (RPG-2024-051) and the EPSRC Mathematical and Foundations of Artificial Intelligence Probabilistic AIHub (EP/Y007174/1). GM acknowledges support from a UKRI AI Turing Acceleration Fellowship (EP/V024868/1).

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

# A  Additional related work: imitation learning

Imitation learning (IL) aims to mimic expert behaviours by learning observation-to-action mappings [54]. behaviour cloning (BC), a prevalent IL method, uses supervised learning but cannot selectively incorporate demonstrations based on estimated performance like our approach. IL's primary challenge is generalising to unseen scenarios with limited demonstrations, stemming from state distribution shifts, demonstrator-learner correspondence problems [55], and i.i.d. assumption violations. Methods like DAgger [56] and Deeply AggreVaTeD [57] address accumulated errors through continuous expert interaction during training—an assumption our method avoids. While effective across domains, IL cannot surpass demonstrator performance and depends heavily on demonstration quality and quantity, unlike our approach which leverages even suboptimal demonstrations while exceeding their performance. Some approaches use RL for post-learning refinement [58, 59], whereas we seamlessly integrate demonstration guidance throughout learning. Crucially, traditional IL methods assume extensive expert demonstrations, while our work targets scenarios with limited, potentially suboptimal demonstration availability.

# B  Missing proofs and further theoretical results

**Proof of Lemma 5.1**  By Assumption (A2), the demonstration samples are i.i.d., so it suffices to show $\mathrm{Var}[Y] \leq \mathrm{Var}[X]$ for a single sample, where $X = \mathbb{1}g$ and $Y = pg$ with $g = \nabla_\phi \|\pi_\phi(s) - a\|^2$. Let $E$ denote the ensemble statistics for a given state-action pair $(s, a)$, which determine both $\mathbb{1}$ and $p$. Applying the law of total variance:

$$\mathrm{Var}(X) = \mathbb{E}[\mathrm{Var}(X \mid E)] + \mathrm{Var}(\mathbb{E}[X \mid E])$$

Since $g$ is fixed given $(s, a)$ and $\phi$, we have:

$$\mathbb{E}[X \mid E] = \mathbb{E}[\mathbb{1}\, g \mid E] = g\, \mathbb{E}[\mathbb{1} \mid E]$$

For SPReD-P, $p$ represents $\mathbb{P}(Q(s, a) > Q(s, \pi_\phi(s)) \mid E)$, which is precisely $\mathbb{E}[\mathbb{1} \mid E]$ under our modeling assumptions. Thus, $\mathbb{E}[X \mid E] = gp = Y$. For the conditional variance term:

$$\mathrm{Var}(X \mid E) = \mathrm{Var}(\mathbb{1}g \mid E) = g^2\, \mathrm{Var}(\mathbb{1} \mid E) = g^2 p(1 - p) \geq 0$$

since $\mathbb{1}$ follows a Bernoulli distribution with parameter $p$ conditional on $E$. Substituting back:

$$\mathrm{Var}(X) = \mathbb{E}[g^2 p(1 - p)] + \mathrm{Var}(Y) \geq \mathrm{Var}(Y)$$

The inequality is strict when $\mathbb{E}[g^2 p(1 - p)] > 0$, which occurs if and only if there exists a non-zero measure set where $g \neq 0$ and $0 < p < 1$ simultaneously. Since $g$ is typically non-zero (by Assumption A1), this simplifies to requiring $\mathbb{P}(0 < p < 1) > 0$.  □

**Remark on SPReD-E.**  Note that the key step in Lemma 5.1 is the identity $p_P = \mathbb{E}\big[\mathbb{1}\{Q_d > Q_\pi\} \mid E\big]$ which lets us write $\mathbb{E}[X \mid E] = g\, p_P$. In the exponential variant $p_E$ is *not* by construction this exact conditional expectation. Nevertheless, $p_E \in [0, 1]$ and it *tracks* the true probability $p_P$ closely shown in Property 5.3 ($p_P - \frac{1}{2} \approx p_E \frac{\beta}{\sigma \sqrt{2\pi}}$). Empirically (Figure 7), SPReD-E therefore exhibits nearly the same reduction in gradient variance as SPReD-P. Extending Lemma 5.1 to cover any smooth, bounded weight $p$ remains an interesting direction for future theoretical work.

**Extension of Property 5.1**  We restate Property 5.1 by separating into the two models: the first for probabilistic advantage weighting and the second for exponential advantage weighting. Property 5.1 is a summary of Property B.1 and Property B.2

**Property B.1** (An exhaustive version of adaptive behaviour for $p_P$)**.**  *Assume $Q(s_d, a_d)$ and $Q(s_d, \pi_\phi(s_d))$ are Gaussian's with variances $\hat{\sigma}_d^2$ and $\hat{\sigma}^2$ respectively. Let $A$ be the difference of their means. We define $p_P = 0$ when $A = 0$ and $\hat{\sigma}^2 + \hat{\sigma}_d^2 = 0$. As the variance varies, our probabilistic advantage weights satisfy:*

*(i)* High-certainty: *If $\hat{\sigma}^2 + \hat{\sigma}_d^2 \to 0$ then $p_P \to \mathbb{1}_{A>0}$ (with $p_P = 0.5$ if $A = 0$).*

*(ii)* High-uncertainty: *If $\hat{\sigma}^2 + \hat{\sigma}_d^2 \to \infty$ then $p_P \to 0.5$.*

*Proof.* Since $p_P = \Phi\left(\frac{A}{\sqrt{\hat{\sigma}_d^2 + \hat{\sigma}^2}}\right)$:

- If $A > 0$ and $\hat{\sigma}_d^2 + \hat{\sigma}^2 \to 0$: $p_P \to \Phi(+\infty) = 1$.

- If $A < 0$ and $\hat{\sigma}_d^2 + \hat{\sigma}^2 \to 0$: $p_P \to \Phi(-\infty) = 0$.

- If $A = 0$ and $\hat{\sigma}_d^2 + \hat{\sigma}^2 > 0$: $p_P = \Phi(0) = \frac{1}{2}$.

- If $\hat{\sigma}_d^2 + \hat{\sigma}^2 \to \infty$: $p_P \to \Phi(0) = \frac{1}{2}$. $\qquad\square$

**Property B.2** (An exhaustive version of adaptive behaviour for $p_E$)**.** *Assume $\beta = \alpha\hat{\beta}$ where $\hat{\beta}$ is the IQR of the mixture model with components $Q(s_d, a_d)$ and $Q(s_d, \pi_\phi(s_d))$. Let $\hat{\sigma}_d^2$ and $\hat{\sigma}^2$ be the variances and $A$ be the difference of means of $Q(s_d, a_d)$ and $Q(s_d, \pi_\phi(s_d))$. Assume (for convenience) that both distributions are continuous and symmetric. We define $p_E = 0$ when $A = 0$ and $\hat{\sigma}^2 + \hat{\sigma}_d^2 = 0$. As the variance varies, our exponential advantage weights satisfy:*

*(i)* High-certainty: *If $\hat{\sigma}^2 + \hat{\sigma}_d^2 \to 0$ then $p_E \to \mathrm{clip}(e^{\frac{1}{\alpha}} - 1, 0, 1)$ when $A > 0$ and $p_E = 0$ when $A \leq 0$.*

*(ii)* High-uncertainty: *If the 4th moments of $Q(s_d, a_d)$ and $Q(s_d, \pi_\phi(s_d))$ scale like $\hat{\sigma}_d^4$ and $\hat{\sigma}^4$ respectively then as $\hat{\sigma}^2 + \hat{\sigma}_d^2 \to \infty$, $p_E \to 0$.*

*Proof.* For notational convenience, let $Q_1 = Q(s_d, a_d)$ and $Q_2 = Q(s_d, \pi_\phi(s_d))$, and for consistency we redefine the variances $\sigma_1 = \hat{\sigma}_d$ and $\sigma_2 = \hat{\sigma}$. We let $m_i$ be the means of $Q_i$ and $Q$ be the mixture model $Q = Q_i$ with probability 0.5 for $i = 1, 2$. We choose $\hat{\beta}$ to be the IQR of $Q$ and $\beta = \alpha\hat{\beta}$ for some $\alpha > 0$ which we fix. In this notation $A = m_1 - m_2$.

*(i) High-certainty limit:* As $\sigma_1 + \sigma_2 \to 0$ we claim that $\hat{\beta} \to A$. By definition, assuming a continuous and symmetric distribution for $Q$, the IQR, $\hat{\beta}$, satisfies $\mathbb{P}(|Q - \hat{m}| \leq \frac{\hat{\beta}}{2}) = 0.5$ (the value of 0.5 does not matter, a larger or smaller quantile bound could be equivalently considered) where $\hat{m} = \frac{m_1 + m_2}{2}$ is the mean of $Q$. Now for any $\delta > 0$ we can apply Chebyshev's inequality to infer

$$\mathbb{P}\left(|Q - \hat{m}| \leq \frac{A}{2} - \delta\right) \leq \frac{1}{2}\mathbb{P}(|Q_1 - m_1| \geq \delta) + \frac{1}{2}\mathbb{P}(|Q_2 - m_2| \geq \delta) \leq \frac{\sigma_1^2}{2\delta^2} + \frac{\sigma_2^2}{2\delta^2} \to 0$$

as $\sigma_1 + \sigma_2 \to 0$. Hence, $\lim_{\sigma_i \to 0} \hat{\beta} > A - 2\delta$ for all $\delta > 0$. In particular, $\lim_{\sigma_i \to 0} \hat{\beta} \geq A$. On the other hand, applying Chebyshev's inequality again implies

$$\mathbb{P}\left(|Q - \hat{m}| < \frac{A}{2} + \delta\right) \geq \frac{1}{2}\mathbb{P}(|Q_1 - m_1| < \delta) + \frac{1}{2}\mathbb{P}(|Q_2 - m_2| < \delta)$$

$$= 1 - \frac{1}{2}\mathbb{P}(|Q_1 - m_1| \geq \delta) - \frac{1}{2}\mathbb{P}(|Q_2 - m_2| \geq \delta)$$

$$\geq 1 - \frac{\sigma_1^2}{2\delta^2} - \frac{\sigma_2^2}{2\delta^2}$$

$$\to 1$$

as $\sigma_1 + \sigma_2 \to 0$ for any $\delta > 0$. This implies $\lim_{\sigma_i \to 0} \hat{\beta} < A + 2\delta$ for all $\delta > 0$. In particular, $\lim_{\sigma_i \to 0} \hat{\beta} \leq A$.

Now for $A > 0$ we have

$$p_E = \mathrm{clip}(e^{\frac{A}{\beta}} - 1, 0, 1) = \mathrm{clip}(e^{\frac{A}{\alpha\hat{\beta}}} - 1, 0, 1) \to \mathrm{clip}(e^{\frac{1}{\alpha}} - 1, 0, 1).$$

For $A \leq 0$ and $\sigma_1 + \sigma_2 > 0$, we have $\frac{A}{\beta} \leq 0$ since $\hat{\beta} > 0$, so $p_E = \mathrm{clip}(e^{\frac{A}{\beta}} - 1, 0, 1) = 0$.

*(ii) High-uncertainty limit*: As either $\sigma_1 \to +\infty$ or $\sigma_2 \to \infty$ we claim $\hat{\beta} \to \infty$. Let $Z = (Q - \hat{m})^2$. Then $\mathbb{E}Z$ is the variance of the mixture model $Q$ which is straightforward to compute as $\sigma^2 = $

$\frac{\sigma_1^2+\sigma_2^2}{2} + \frac{A^2}{4}$. If $M$ is an upper bound on the 4th moment of $Q_1$ and $Q_2$ then a direct computation with Jensen's inequality gives the bound

$$\mathbb{E}Z^2 \leq M + 4\hat{m}M^{\frac{3}{4}} + 6\hat{m}^2 M^{\frac{1}{2}} + 3\hat{m}^4 \leq CM \leq \tilde{C}\sigma^4$$

for some $C, \tilde{C} > 0$. By the Paley–Zygmund inequality,

$$\mathbb{P}\left(|Q - \hat{m}| \geq R\right) = \mathbb{P}\left(Z \geq R^2\right) \geq \left(1 - \frac{R^2}{\mathbb{E}Z}\right)^2 \frac{(\mathbb{E}Z)^2}{\mathbb{E}Z^2} \geq \frac{1}{\tilde{C}}\left(1 - \frac{R^2}{\sigma^2}\right)^2$$

for any $R > 0$. If we choose $R^2 = \sigma^2\left(1 + \sqrt{\frac{3\tilde{C}}{4}}\right)$ then we have

$$\mathbb{P}\left(|Q - \hat{m}| \geq R\right) \geq 0.75.$$

It follows that $\hat{\beta} \geq R$. As $R \to \infty$ then $\hat{\beta} \to \infty$.

It is straightforward to now conclude that

$$p_E = \text{clip}(e^{\frac{A}{\beta}} - 1, 0, 1) \to \text{clip}(0, 0, 1) = 0. \qquad \square$$

**Remark.** *For $A < 0$ and $\sigma_1 + \sigma_2 = 0$, $\hat{\beta} = m_2 - m_1$, so*

$$p_E = \text{clip}\left(e^{\frac{A}{\beta}} - 1, 0, 1\right) = \text{clip}\left(e^{\frac{A}{\alpha\beta}} - 1, 0, 1\right) = \text{clip}\left(e^{-\frac{1}{\alpha}} - 1, 0, 1\right) = 0.$$

*For $A = 0$ and $\sigma_1 + \sigma_2 = 0$, we define $p_E = 0$ for consistency.*

**Remark.** *When $A = 0$ and $\hat{\sigma}_d^2 + \hat{\sigma}^2 = 0$, $Q(s_d, a_d) = Q(s_d, \pi_\phi(s_d))$ almost surely, which leads to $p_P = 0$ if using a strict inequality and $p_P = 1$ if using a non-strict inequality in the definition of $p_P$, neither desirable. Hence, we define $p_P = \frac{1}{2}$.*

**Proof of Property 5.2** Fix $(s_d, a_d)$ with

$$\Delta Q^* := Q^*(s_d, \pi^*(s_d)) - Q^*(s_d, a_d) > 0.$$

Let

$$A_t = \hat{Q}(s_d, a_d) - \hat{Q}(s_d, \pi_{\phi_t}(s_d)).$$

By (A3), there exists a sequence $\varepsilon_t \to 0$ as $t \to \infty$ such that

$$\sup_a |\hat{Q}_t(s_d, a) - Q^*(s_d, a)| \leq \varepsilon_t.$$

In particular,

$$|\hat{Q}_t(s_d, a_d) - Q^*(s_d, a_d)| \leq \varepsilon_t \quad \text{and} \quad |\hat{Q}_t(s_d, \pi_{\phi_t}(s_d)) - Q^*(s_d, \pi_{\phi_t}(s_d))| \leq \varepsilon_t.$$

Since the policy improves, for any $\delta > 0$ there is $T$ with

$$Q^*(s_d, \pi_{\phi_t}(s_d)) \geq Q^*(s_d, \pi^*(s_d)) - \delta \quad \forall t \geq T.$$

Hence, for $t \geq T$,

$$A_t \leq \left(Q^*(s_d, a_d) + \varepsilon_t\right) - \left(Q^*(s_d, \pi^*(s_d)) - \delta - \varepsilon_t\right) = -\Delta Q^* + 2\varepsilon_t + \delta.$$

Choosing $T$ large enough that $2\varepsilon_t + \delta < \Delta Q^*$ for all $t \geq T$ implies $A_t < 0$. By the high-certainty limit of Property 5.1, as the ensemble variance vanishes we get

$$p_P(s_d, a_d) \to \mathbb{1}_{A_t > 0} = 0, \quad p_E(s_d, a_d) = 0. \qquad \square$$

**Proof of Property 5.3**

*Proof.* For the standard normal CDF,

$$\Phi(x) = \frac{1}{2} + \frac{x}{\sqrt{2\pi}} - \frac{x^3}{6\sqrt{2\pi}} + \mathcal{O}(x^5)$$

Setting $x = A/\sigma$ yields the expansion of $p_P = \Phi(A/\sigma)$. For $p_E = \exp(A/\beta) - 1$ (where clipping is inactive when $|A|/\beta \ll 1$), we use the power series of the exponential at 0:

$$\exp(y) = 1 + y + \frac{y^2}{2} + \frac{y^3}{6} + \mathcal{O}(y^4)$$

Thus $p_E = \exp(A/\beta) - 1 = \frac{A}{\beta} + \frac{A^2}{2\beta^2} + \frac{A^3}{6\beta^3} + \mathcal{O}((A/\beta)^4)$. Comparing the linear terms of both expansions, we see that $p_P - \frac{1}{2} \approx \frac{A}{\sigma\sqrt{2\pi}}$ and $p_E \approx \frac{A}{\beta}$. These terms match when $\beta = \sigma\sqrt{2\pi}$. $\qquad \square$

**Remark.** *Empirically, our ensemble Q-values are near-Gaussian, and also the difference between two independent distributions, where* $\mathrm{IQR} \approx 1.35\,\sigma$. *Given the theoretical relationship* $\beta = \sigma\sqrt{2\pi}$, *and substituting this into* $\beta = \alpha \cdot \mathrm{IQR}$, *we can determine a principled starting point for* $\beta$ *and* $\alpha$. *While* $\alpha$ *requires tuning for specific task characteristics, the theoretical relationship provides a meaningful baseline that ensures proper uncertainty scaling across environments.*

## C  Algorithms and implementation details

**Algorithm overview**    SPReD incorporates uncertainty quantification to enable adaptive demonstration utilisation through a principled ensemble-based approach. We maintain an ensemble of $m$ critic networks ($\theta_i$), each with a corresponding target network, alongside a standard actor network ($\phi$) with its target network. The algorithm maintains two separate buffers: a standard experience replay buffer $\mathcal{B}$ and a demonstration buffer $\mathcal{B}_D$ containing pre-collected demonstration transitions.

---

**Algorithm 1** Reinforcement Learning with Smooth Policy regularisation from Demonstrations

---

1: Initialise critic networks $\theta_i$, actor network $\phi$ and their target networks $\theta_i' \leftarrow \theta_i$, $\phi' \leftarrow \phi$, where $i = 1, 2, ..., m$ and $m$ is the ensemble size
2: Initialise replay buffer $\mathcal{B} = \emptyset$ and demonstration buffer $\mathcal{B}_D$ with transitions in demonstrations $(s_d, a_d, r_d, s_d', g_d)$
3: **for** episode $e = 1$ to $M$ **do**
4:     **for** $t = 1$ to $T$ **do**
5:         Execute action $a \sim \pi_\phi(s) + \epsilon$ with exploration noise $\epsilon \sim \mathrm{clip}(\mathcal{N}(0, \sigma), -c, c)$ where $c$ is the maximum action. Observe reward $r$ and new state $s'$
6:         Store the transition $(s, a, r, s', g)$ in $\mathcal{B}$
7:         Update state $s \leftarrow s'$
8:     **end for**
9:     **for** $t = 1$ to $T$ **do**
10:         Store the transition $(s, a, r, s', g_a)$ in $\mathcal{B}$ where $g_a$ is the actual achieved goal in this episode
11:     **end for**
12:     **for** iteration $l = 1$ to $k$ **do**
13:         Sample mini-batch of size $N_R$ from $\mathcal{B}$ and mini-batch of size $N_D$ from $\mathcal{B}_D$.
14:         Sample two critics uniformly from the ensembles
15:         Update all critic networks by minimizing

$$\mathcal{L}(\theta_i) = \mathbb{E}_{(s,a) \sim \mathcal{B}, \mathcal{B}_D}(r + \gamma \min_{i=1,2} Q_{\theta_i'}(s', \tilde{a}) - Q_{\theta_i}(s, a))^2$$

16:         where $\tilde{a} = \pi_{\phi'}(s') + \epsilon$, $\epsilon \sim \mathrm{clip}(\mathcal{N}(0, \sigma'), -c', c')$
17:         **if** $l \bmod d = 0$ **then**
18:             Update the actor network by minimizing Equation 1
19:             Update target networks:
20:             $\theta_i' \leftarrow \tau\theta_i + (1 - \tau)\theta_i'$
21:             $\phi' \leftarrow \tau\phi + (1 - \tau)\phi'$
22:         **end if**
23:     **end for**
24: **end for**

---

Our Algorithm 1 operates in two main phases. During environment interaction (lines 4-8), the agent follows its current policy with added exploration noise bounded to the action space to collect experiences. Following the HER approach [41], we augment collected transitions by storing them again with actually achieved goals (lines 9-11), enabling learning from unsuccessful episodes in sparse-reward environments. The learning process (lines 12-21) integrates several key components:

- **Coordinated sampling**: Each update draws transitions from both experience and demonstration buffers with fixed proportions, ensuring consistent demonstration influence throughout training.

- **Ensemble-based target computation**: We randomly select two critics from the ensemble to compute target values, following the REDQ approach [53] to mitigate overestimation bias while maintaining computational efficiency.

- **Uncertainty-aware policy updates**: The actor loss combines standard deterministic policy gradient with our uncertainty-weighted behaviour cloning loss that smoothly regularises the policy.

**Weighting mechanisms** During the actor update, we compute the weight $p$ for each demonstration using either SPReD-P or SPReD-E. For SPReD-P, we compute the probability as described in Section 4, which naturally produces values in the $[0, 1]$ range through the CDF. For SPReD-E, we practically take

$$\beta = \alpha \cdot \frac{1}{2}\big[\text{IQR}\{Q_i(s_d, a_d)\}_{i=1}^m + \text{IQR}\{Q_i(s_d, \pi(s_d))\}_{i=1}^m\big]$$

and apply the truncation to the basic exponential form:

$$p_E = \text{clip}(e^{A/\beta} - 1, 0, 1)$$

This truncation: (1) ensures $p_E = 0$ for negative advantages, preventing imitation of demonstrably inferior actions; (2) creates a smooth exponential ramp for modest positive advantages; and (3) caps the weight at $p_E = 1$ when $A/\beta \geq \ln 2$, providing full imitation only for clearly superior demonstrations.

**Remark.** *Both our choice of $\beta$ and $\beta^* = \sigma\sqrt{2\pi}$, which connects two variants of our method, measure the uncertainty of the advantage. The choice of $\beta$ is not sensitive, and $\beta$ provides proportional bounds for $\beta^*$. With near-Gaussian Q-value distributions, our $\beta \approx \frac{1.35(\hat{\sigma}+\hat{\sigma}_d)}{2}$. By the Cauchy-Schwarz inequality,*

$$\pi(\hat{\sigma} + \hat{\sigma}_d)^2 \leq (\beta^*)^2 = 2\pi(\hat{\sigma}^2 + \hat{\sigma}_d^2) \leq 2\pi(\hat{\sigma} + \hat{\sigma}_d)^2.$$

*Then we have $\frac{2\sqrt{\pi}}{1.35}\beta \leq \beta^* \leq \frac{2\sqrt{2\pi}}{1.35}\beta$ since $\beta$ and $\beta^*$ are non-negative.*

**Computational considerations** Despite its theoretical sophistication, SPReD introduces minimal computational overhead. The entire probability or advantage calculation and weighting process requires only a few simple operations beyond standard TD3, with the ensemble critics serving dual purposes of target value computation and uncertainty estimation. The actor update occurs less frequently (every $d$ steps) to allow critic networks to stabilise between policy updates. Our implementation maintains the same overall complexity as standard RL algorithms while gaining the benefits of uncertainty-aware demonstration utilisation. The practical computational cost of our method and all baselines we compare are presented in Figure 4. SPReD requires $\approx 2.6$ hours per 4M environment steps, nearly identical to TD3 (2 critics), while RLPD (also 10 critics) requires $\approx 4.8$ hours. SPReD processes $\approx 427$ environment steps/second, compared to TD3's $\approx 444$ steps/second—only a $3.8\%$ decrease despite using $5\times$ more critics. Given that SPReD achieves up to $14\times$ better success rates than baselines on complex tasks, this $< 4\%$ throughput cost is well justified. Our SPReD method requires almost the same computational resources as the standard RL algorithm, while a single running of RLPD takes nearly double time.

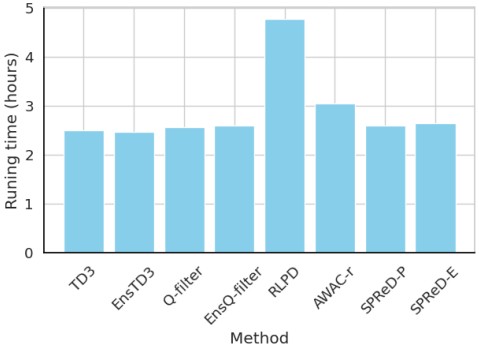

Figure 4: Computational cost for the individual experimental runs for FetchPickAndPlace with 4e6 steps.

**Network architecture and hyperparameters** We implement our approach using hyperparameters consistent with prior work [6]: mini-batch sizes $N_R = 1024$ and $N_D = 128$ for experience and demonstration buffers respectively, discount factor $\gamma = 0.98$, and loss weights $\lambda_1 = 10^{-3}$ and $\lambda_2 = \frac{1}{128}$. Both actor and critic networks employ identical architectures consisting of two hidden layers with 256 neurons each and ReLU activations. The actor's output layer uses a tanh activation to bound actions within the environment's range. We use the Adam optimizer [60] with learning rate $10^{-3}$ for all networks.

For SPReD-E, the scaling constant $\alpha$ was set to 10 based on preliminary experiments. This value provides sufficient caution with uncertain estimates while still allowing clear advantages to receive significant imitation weights. The ablation test is presented in Appendix G.

State and goal observations are normalised before being processed by the networks:

$$\text{Normalised value} = \text{clip}\left(\frac{\text{clip(original value}, -200, 200) - \mu}{\sigma + 10^{-6}}, -5, 5\right) \tag{2}$$

where $\mu$ and $\sigma$ represent the running mean and standard deviation, updated at each step using Welford's online algorithm [61]. For methods using demonstrations, these statistics are initialised using the demonstration data. The small constant ($10^{-6}$) prevents division by zero.

For exploration, we employ clipped Gaussian noise with $\sigma = 0.1$. Following standard TD3 implementation [38], we add smoothing noise $\text{clip}(\mathcal{N}(0, 0.2), -0.5, 0.5)$ to actions during critic updates to enhance Q-function smoothness across similar actions.

We maintain a replay buffer capacity of $10^6$ transitions and begin training after collecting $10N_R$ initial transitions. Critic networks are updated twice per iteration while the actor network is updated once, with target networks updated via Polyak averaging using $\tau = 10^{-3}$. Our ensemble consists of 10 independent critic networks with ablation test presented in Appendix G. Policy performance is evaluated regularly over 25 test episodes without exploration noise. All the experiments were performed with a single GeForce GTX 3090 GPU and an Intel Core i9-11900K CPU at 3.50GHz.

## D  Further environment details

We evaluate our approach on eight robotics tasks implemented in the OpenAI Gym framework [49], simulated using the MuJoCo physics engine [50]. These environments feature sparse rewards and multi-goal structures, providing an ideal testbed for demonstration-based learning approaches.

**Fetch robotic arm tasks** The Fetch environment employs a 7-DoF robotic arm with a parallel gripper for manipulation tasks of increasing complexity:

- **FetchPush**: Moving objects to target positions on a tabletop
- **FetchSlide**: Striking objects toward targets beyond the arm's reach
- **FetchPickAndPlace**: Lifting and positioning objects in 3D space
- **FetchStack2** and **FetchStack3**: Precisely arranging multiple blocks in specified configurations

In these environments, the action space is 4-dimensional, with the first three dimensions controlling the gripper's movement and the fourth dimension controlling gripper opening/closing. Observations are 25-dimensional, containing position and velocity information for both the gripper and manipulated objects, and goals are specified as 3-dimensional target positions in Cartesian coordinates for standard tasks. Additional dimensions are included for stacking tasks to accommodate multiple objects.

**Shadow dexterous hand tasks** The Shadow Hand environment presents significantly more complex control challenges:

- **ManipulateBlock**: Manipulating a cube to a target orientation
- **ManipulateEgg**: Orienting an egg-shaped object
- **ManipulatePen**: Precisely controlling a pen-shaped object

Goals in these environments are specified as 7-dimensional vectors representing target positions (3D Cartesian coordinates) and orientations (quaternions). The action space is 20-dimensional, corresponding to the absolute angular positions of the hand's actuated joints. Observations are 61-dimensional, containing comprehensive kinematic information about both the hand and the manipulated object.

All environments employ sparse binary rewards, with agents receiving 0 when successfully achieving goals (within a specified tolerance) and -1 otherwise. For the stacking tasks, an additional reward of +1 is provided when the gripper moves away from the blocks after successful completion, encouraging proper task termination.

## E  Demonstration data quality

We collect demonstrations with varying levels of quality to evaluate the robustness of our approach across different conditions. For each environment, we use 100 demonstration episodes, with the exception of FetchStack3 where we use 1000 episodes due to its greater complexity.

Table 2: Categorisation of environments by demonstration quality used in main results reported by Table 1 and Figure 1.

| Quality Level | Success Rate | Environments |
| --- | --- | --- |
| Expert | 0.86-1.00 | FetchStack2 (1.00), FetchStack3 (1.00), ManipulateBlock (0.86) |
| Moderate | 0.49-0.53 | FetchSlide (0.53), FetchPickAndPlace (0.49) |
| Low | 0.20-0.39 | ManipulateEgg (0.39), ManipulatePen (0.37), FetchPush (0.20) |

For the challenging stacking tasks (FetchStack2 and FetchStack3), we utilise expert demonstrations from policies developed by Lanier et al. [51], achieving perfect success rates (1.0). The ManipulateBlock environment also features high-quality demonstrations with a success rate of 0.86.

For the remaining environments, we generate demonstrations of varying quality using policies trained with EnsTD3+HER and introducing controlled levels of noise. FetchSlide (0.53) and FetchPickAnd-Place (0.49) use moderate-quality demonstrations, while ManipulateEgg (0.39), ManipulatePen (0.37), and FetchPush (0.20) employ lower-quality demonstrations. Table 2 categorises these environments by demonstration quality level.

For experiments explicitly analyzing sensitivity to demonstration quality (Figure 2 and Figure 10), we generate three distinct quality levels for each environment:

1. **Expert**: Demonstrations collected from well-trained policies with minimal added noise.
2. **Suboptimal**: Generated by adding moderate Gaussian noise to expert actions.
3. **Severely suboptimal**: Created with substantial noise that significantly degrades demonstration quality.

This systematic variation enables us to precisely characterise how each algorithm's performance scales with demonstration quality, isolating this factor from other variables. For the extreme test case, we also evaluate performance when provided with 99% random trajectories mixed with just 1% expert demonstrations.

The deliberate inclusion of diverse demonstration qualities reflects real-world scenarios where perfect demonstrations may be unavailable or prohibitively expensive to collect. An algorithm's ability to extract useful information even from imperfect demonstrations is particularly important for practical applications.

## F  Empirical evidence to support methods

**Uncertainty measures**   As we mention in Section 2, there are different methods of the uncertainty measure. We investigate dropout-based uncertainty as an alternative to our ensemble approach. Contrary to the intuition that dropout might reduce overhead, our experiments show it actually increases computational cost while degrading performance:

Table 3: Running time and success rate for SPReD-P with different uncertainty measures in Fetch-PickAndPlace. The ensemble size for ensemble method is 10. For dropout method, the dropout rate is 0.1, and there are 500 forward passes per critic (2 critics in TD3).

| Method | Time (4M steps) | Success rate (1M steps) |
|---|---|---|
| Ensemble | 2.6h | $0.832 \pm 0.111$ |
| Dropout | 20.3h | $0.600 \pm 0.057$ |

The dropout approach is 8× slower due to requiring 1000 stochastic forward passes (500 × 2) to estimate uncertainty, while our ensemble uses a single vectorised pass through 10 critics in parallel on GPU. Although both methods eventually learn an expert policy within 4M steps, dropout shows worse sample efficiency (28% drop), likely due to: (1) noisier uncertainty estimates, (2) overconfident predictions on unseen data [62], and (3) computational bottlenecks from repeated passes that hinder efficient batch processing.

These results are consistent with prior findings [62] that ensembles yield better uncertainty estimates than dropout, and modern GPUs enable highly efficient ensemble parallelisation.

Bootstrapping presents other computational challenges: (1) separate data samples per model prevent batch-sharing, (2) models process different minibatches, blocking parallelisation, and (3) storing multiple bootstrap samples increases memory demands. In contrast, our ensemble processes the same batch across all critics via a single vectorised operation, achieving near-linear speedup. Moreover, Bootstrapped DQN [30] supports this strategy, suggesting that diversity from random initialisations of deep NN eliminates the need of explicit data bootstrapping.

These results confirm that vectorised ensembles offer the best balance between uncertainty estimation quality and computational efficiency.

**Locomotion tasks** We also evaluate our methods for relatively easy OpenAI Gym locomotion tasks [52] with dense rewards, where goals and HER are excluded from the algorithm. The tasks are to move the following robots in the forward direction:

- **Hopper**: a two-dimensional one-legged figure
- **HalfCheetah**: a two-dimensional robot with 9 body parts and 8 joints
- **Walker2d**: a two-dimensional bipedal robot
- **Ant**: a three-dimensional quadruped robot
- **Humanoid**: a three-dimensional bipedal robot which simulates a human

According to the initial sample efficiency at 200K steps shown in Table 4, SPReD method surpasses all variants of AWAC for HalfCheetah and Ant, and is particularly outstanding on HalfCheetah (SPReD-P gains 12% improvement from EnsQ-filter and 72% improvement from Q-filter). AWAC is competitive for other tasks with near-expert demonstrations, and RLPD has remarkable sample efficiency initially. However, the learning scores of AWAC and RLPD are asymptotically lower than SPReD for all locomotion tasks we tested, which is visualised by learning curves in Figure 5. Even in setting with dense rewards, our SPReD method is the robustest with relatively high sample efficiency, consistent improvement and the best converging performance, confirming that our uncertainty-aware approach effectively transfers across different continuous control domains.

Table 4: Average score (with standard deviation) over 5 seeds after 200K interactions for locomotion tasks. The highlighted results lie between the mean of the best performer and one standard deviation below it (i.e., if the best result is $\mu \pm \sigma$, all values $\geq \mu - \sigma$ are bold). The scores of demonstrations range from 2500-7000.

| Environment | Methods | | | | | | | | | |
|---|---|---|---|---|---|---|---|---|---|---|
| | TD3 | EnsTD3 | Q-filter | EnsQ-filter | RLPD | AWAC | AWAC-p | AWAC-r | SPReD-P | SPReD-E |
| Hopper | $1468 \pm 849$ | $2144 \pm 966$ | $2818 \pm 438$ | $2930 \pm 546$ | $2744 \pm 580$ | $2855 \pm 707$ | $\mathbf{3228 \pm 50}$ | $2461 \pm 1042$ | $\mathbf{3246 \pm 18}$ | $2740 \pm 344$ |
| HalfCheetah | $3728 \pm 419$ | $3775 \pm 1756$ | $4671 \pm 760$ | $7188 \pm 934$ | $6721 \pm 3542$ | $4835 \pm 854$ | $5150 \pm 801$ | $4209 \pm 1069$ | $\mathbf{8060 \pm 171}$ | $7336 \pm 601$ |
| Walker2d | $1861 \pm 1159$ | $2334 \pm 772$ | $1485 \pm 589$ | $3576 \pm 528$ | $\mathbf{4519 \pm 154}$ | $3987 \pm 108$ | $3343 \pm 1696$ | $4005 \pm 433$ | $3351 \pm 842$ | $2403 \pm 1154$ |
| Ant | $2028 \pm 643$ | $3496 \pm 669$ | $2478 \pm 960$ | $5862 \pm 292$ | $\mathbf{6068 \pm 42}$ | $3470 \pm 1325$ | $-4 \pm 1776$ | $3465 \pm 987$ | $5636 \pm 496$ | $5779 \pm 318$ |
| Humanoid | $853 \pm 242$ | $3588 \pm 2039$ | $4489 \pm 469$ | $5066 \pm 143$ | $4701 \pm 432$ | $\mathbf{5261 \pm 32}$ | $4592 \pm 545$ | $\mathbf{5293 \pm 52}$ | $4920 \pm 542$ | $4806 \pm 557$ |

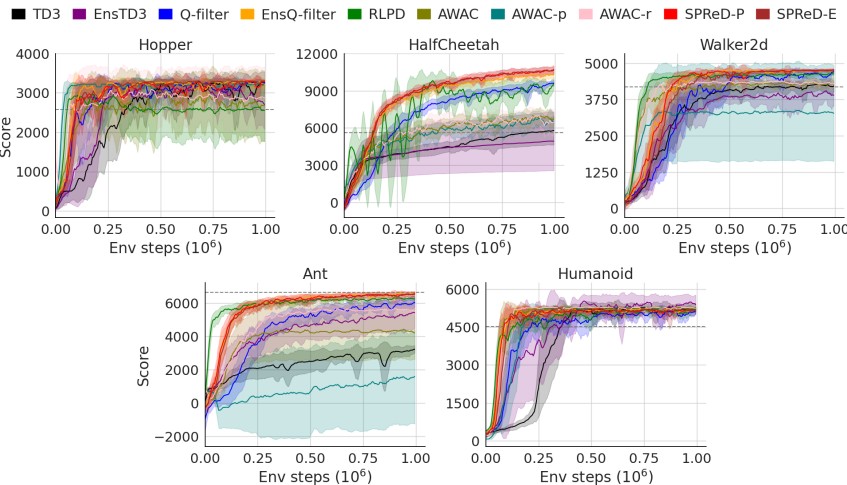

Figure 5: Performance comparison across five locomotion tasks. Horizontal dashed lines indicate the scores of the demonstrations used for training. Our SPReD methods (red and brown) consistently outperform baselines across different tasks.

AWAC's poor performance on both locomotion and manipulation domains stems from a fundamental mismatch with our problem setting: AWAC assumes large offline datasets ( 1M transitions) for effective pretraining, but we have only 5K demonstration transitions. Without sufficient pretraining data, and since demonstrations are quickly diluted in the replay buffer, their impact fades early, effectively reducing AWAC to standard RL. Moreover, AWAC's advantage weighting assumes the offline data covers a substantial portion of the state space, which doesn't hold with sparse demonstrations.

**Gaussian assumption** Our SPReD-P method relies on the assumption that Q-value estimates across the ensemble follow a Gaussian distribution. To validate this assumption, we compare the performance of our Gaussian approach against two nonparametric alternatives that make no distributional assumptions: (1) Nonpara_pairwise, which randomly pairs critic networks and makes pairwise comparisons, and (2) Nonpara_cross, which performs all possible cross-comparisons between critics ($10 \times 10$ pairs).

As demonstrated in Figure 6, the Gaussian approximation consistently outperforms both nonparametric methods in the FetchPickAndPlace environment. These results validate our modeling choice, suggesting that the Gaussian approximation effectively captures the underlying uncertainty while

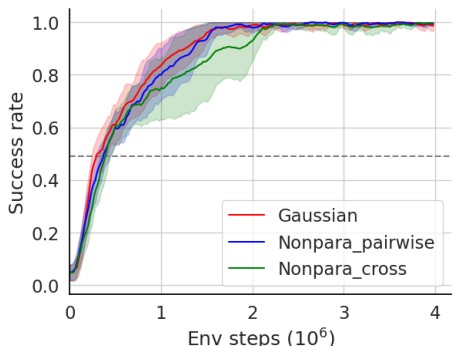

Figure 6: Comparison between Gaussian and nonparametric methods in FetchPickAndPlace. Gaussian approximates two distributions of Q-values as Gaussian distributions using the sample mean and variance. Nonpara_pairwise pairwise compares 10 pairs of Q-values, while Nonpara_cross crosswise compares $10 \times 10$ pairs of Q-values. The dashed line shows the success rate of demonstrations.

providing computational advantages over nonparametric approaches. The superior performance likely stems from the Gaussian model's ability to leverage the entire ensemble's information in a statistically efficient manner, whereas the nonparametric approaches may suffer from higher variance in their comparisons.

**Variance reduction** Through weighted BC, the gradient variance of policy updates is significantly reduced in our SPReD method as proved by Lemma 5.1. Figure 7 confirms our theoretical prediction and establishes the benefit of smooth policy regularisation from demonstrations.

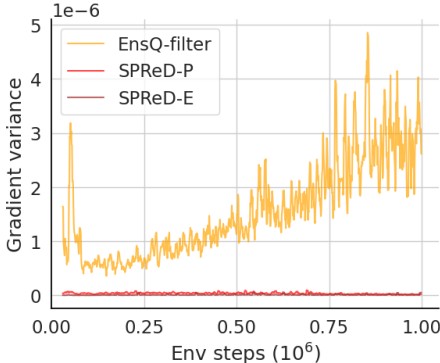

Figure 7: Empirical gradient variance of actor updates in the FetchPush environment, demonstrating that two variants of our SPReD method significantly reduce policy update variance compared to binary imitation decisions.

**Adaptive mechanisms** Figure 8 provides insight into the adaptive mechanism behind the robustness of our SPReD method with various demonstration qualities. Both probabilistic and exponential variants progressively reduce the influence of these extremely suboptimal demonstrations (with only a 0.06 success rate), effectively eliminating their impact on policy updates. Examining SPReD-P's weighting mechanism in detail (Figure 8a), we observe three distinct phases which coincide with our theoretical expectation in Property B.1: (1) an initial uncertainty phase where most weights cluster around 0.5, reflecting limited confidence in Q-value comparisons; (2) a transition phase around one million interactions where the policy improves and weights begin polarizing; and (3) a final

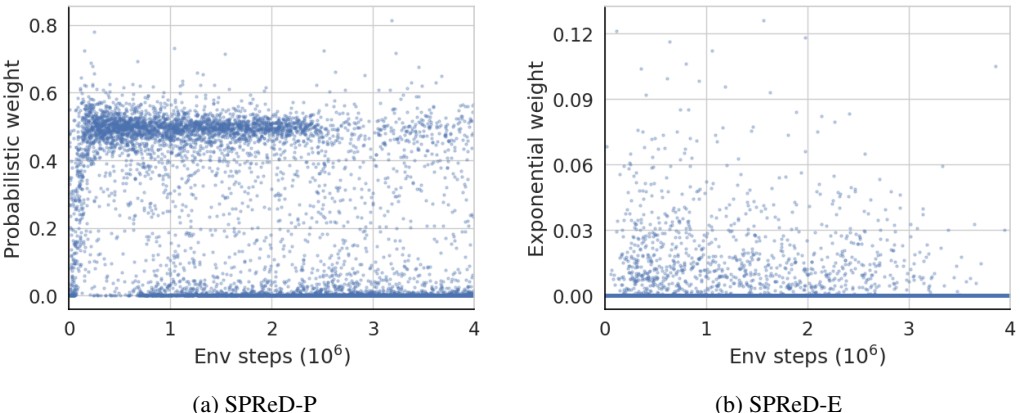

(a) SPReD-P                  (b) SPReD-E

Figure 8: Evolution of behaviour cloning weights in the FetchPickAndPlace environment with severely suboptimal demonstrations (99% random trajectories). SPReD-P (left) shows three distinct phases: initial uncertainty (weights ≈ 0.5), transition (polarizing weights), and final expert policy (most weights → 0). SPReD-E (right) displays a similar trend with generally lower weight magnitudes for generally inferior demonstrations. Both methods automatically reduce the influence of poor demonstrations as learning progresses, demonstrating the adaptive nature of the uncertainty-aware weighting mechanisms.

phase where the vast majority of demonstration weights approach zero, with only genuinely superior demonstration actions retaining influence. SPReD-E (Figure 8b) shows similar qualitative behaviour, though with generally lower weight magnitudes due to its exponential scaling and conforms the theoretical results in Property B.2 with a pessimistic normalisation $\alpha$.

**Noisy rewards**   We also evaluate the robustness of our methods to noisy rewards. We consider two types of noisy rewards: with probability of 0.1, flipping rewards of -1 (failure) to be 0 (success) or adding Gaussian noise to the rewards of -1. Note that reward computation function in HER is still accurate and not affected.

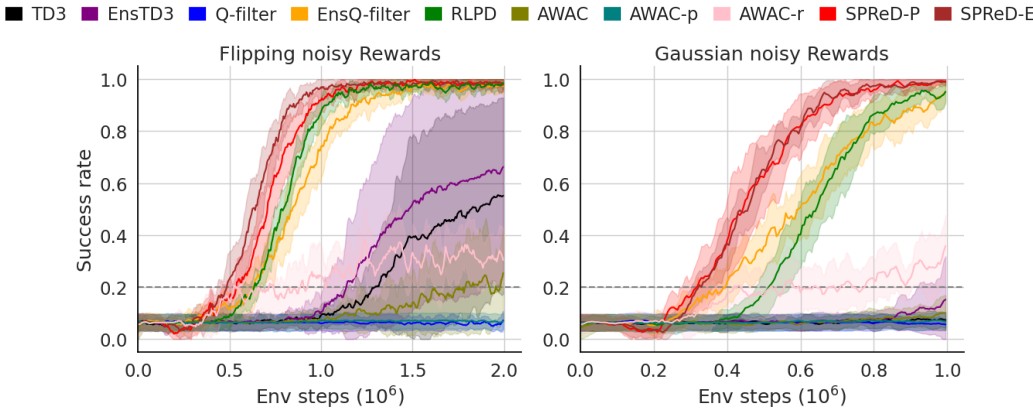

Figure 9: The robustness to noisy rewards in FetchPush. The success rate of demonstrations is shown as dashed lines.

With noisy rewards, the efficient utilisation of demonstrations becomes more crucial. With either kind of noisy rewards, the standard Q-filter fails to learn the policy due to the disturbed Q-values. However, our SPReD methods remain the best among all baselines, benefiting from smooth and uncertainty-aware regularisations.

## G   Ablation study: impact of ensemble size and normalisation constant of SPReD-E on performance

**Additional experiments of demonstration quality**   The experiments in FetchPush exhibit similar trends as in FetchPickAndPlace, confirming the robustness of our methods to demonstration quality.

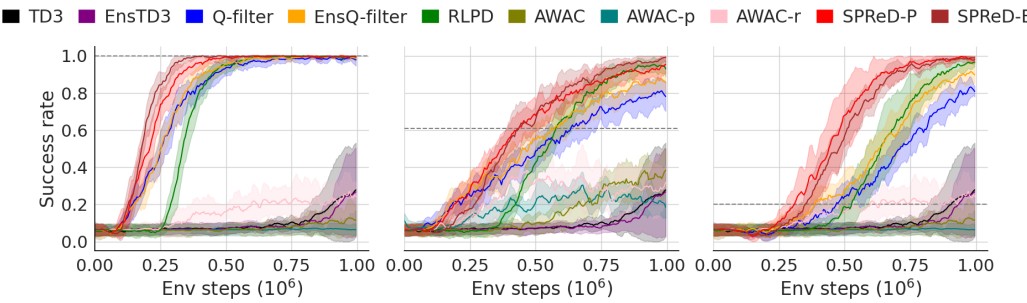

Figure 10: Effect of demonstration quality in FetchPush. The demonstrations are expert, suboptimal and severely suboptimal from left to right with success rates shown as dashed lines.

**Ensemble size**   We systematically evaluate the effect of ensemble size on learning performance by varying it from 2 to 30 critics across multiple environments. Our analysis reveals task-dependent sensitivity to ensemble size. As shown in Figure 11, performance in the FetchPickAndPlace envi-

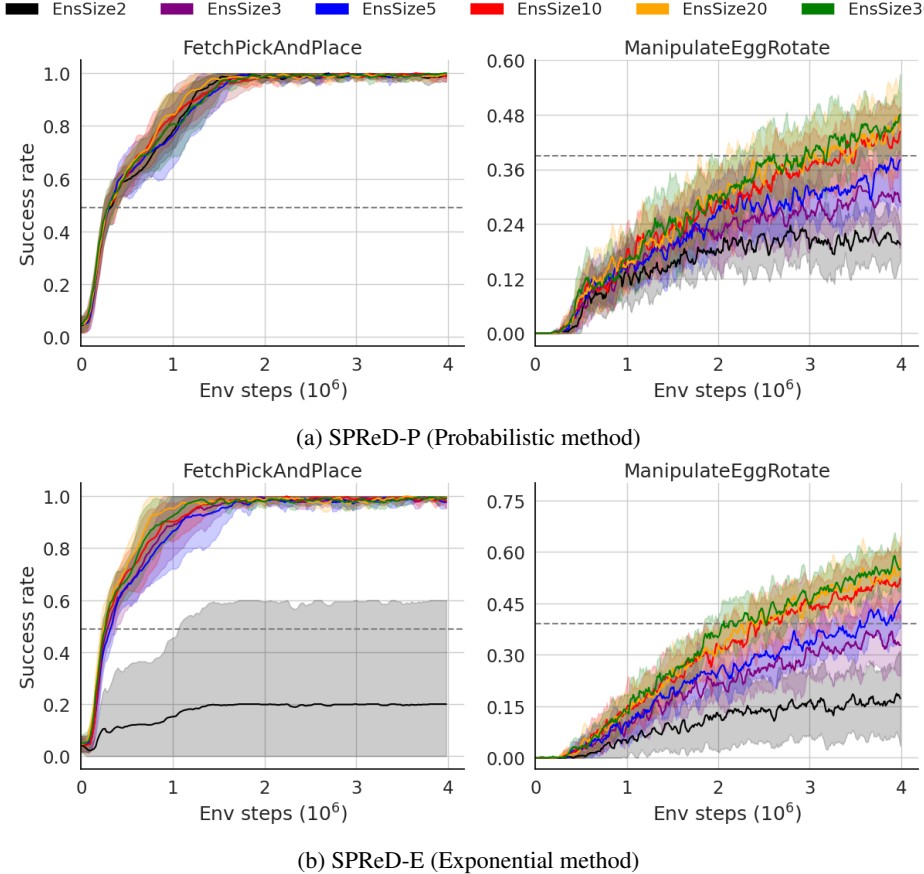

(a) SPReD-P (Probabilistic method)

(b) SPReD-E (Exponential method)

Figure 11: Effect of ensemble size on learning performance. Left: FetchPickAndPlace environment shows minimal sensitivity to ensemble size for both methods. Right: ManipulateEgg environment demonstrates improved performance with larger ensembles, with diminishing returns beyond size 10. Ensemble sizes tested: 2, 3, 5, 10, 20, and 30.

ronment (left) remains relatively consistent across different ensemble sizes for both SPReD variants, suggesting that even small ensembles ($> 2$) capture sufficient uncertainty information for this task.

In contrast, the more complex ManipulateEgg manipulation task (right) shows a clearer performance improvement with increasing ensemble size. This suggests that more challenging control tasks with higher-dimensional action spaces benefit from the improved uncertainty estimates provided by larger ensembles. However, the performance gains diminish noticeably beyond an ensemble size of 10, with minimal additional improvement at sizes 20 and 30 despite the substantial increase in computational cost.

Based on this analysis and computational efficiency considerations, we select an ensemble size of 10 for our main experiments. This choice provides a favorable trade-off between performance and computational requirements, and maintains consistency with the RLPD baseline which also uses 10 critics. Further scaling the ensemble provides diminishing returns that do not justify the additional computational overhead for most practical applications.

**Isolated contribution** Since SPReD introduces both continuous weights and ensemble-based uncertainty estimation, it is important to understand their individual and combined effects. We conduct a systematic ablation study to isolate these contributions. By varying ensemble size (2 vs. 10) and regularisation type (binary vs. continuous), we can assess each component's impact (with success rates at 1M steps).

From results in Table 5, we have the following key findings. First, continuous regularisation is the primary driver. Even with a minimal ensemble (size 2), SPReD-P improves over Q-filter by 28%

Table 5: The success rates of different methods in FetchPickAndPlace at 1M steps to assess isolated contributions of ensemble and continuous weights.

| Q-filter | EnsQ-filter | SPReD-P (size 2) | SPReD-E (size 2) | SPReD-P (size 10) | SPReD-E (size 10) |
|---|---|---|---|---|---|
| $0.608 \pm 0.047$ | $0.688 \pm 0.069$ | $0.776 \pm 0.070$ | $0.152 \pm 0.304$ | $0.832 \pm 0.111$ | $0.888 \pm 0.064$ |

(0.776 vs 0.608). This demonstrates that smooth, uncertainty-proportional weights are fundamentally better than binary decisions, regardless of ensemble size. Then ensembles add complementary value. Expanding from 2 to 10 critics further boosts performance for both SPReD variants. However, the gains are method-specific—SPReD-P shows modest improvement (+7%, 0.776 vs 0.832) while SPReD-E shows dramatic improvement (+484%, 0.152 vs 0.888 ). This differential benefit reveals an important insight that the methods have different uncertainty requirements. SPReD-E's exponential weighting critically depends on accurate uncertainty estimates through $\beta$. With only 2 critics, the IQR calculation is noisy, leading to suboptimal scaling. In contrast, SPReD-P's probabilistic approach is inherently more robust to limited ensemble sizes. The ablation confirms that while both components contribute independently, they work synergistically. Continuous regularisation provides the foundation for better learning, while larger ensembles enable more precise uncertainty quantification—particularly crucial for SPReD-E's exponential scaling mechanism. This validates our design decision to combine both innovations rather than pursuing either in isolation.

**Normalisation constant** The theoretical scaling $\beta = \sigma\sqrt{2\pi}$ in Property 5.3 builds a connection with our probabilistic advantage weighting method SPReD-P, but there is no guarantee that it works best. While both $\sigma$ and IQR measure Q-value distribution spread, the IQR-based approach provides better empirical performance, particularly in more challenging tasks (17% improvement in FetchPickAndPlace with $\alpha = 10$ after 1M steps) as shown in Figure 12. The advantage is even more pronounced with lower-quality demonstrations (44% higher success rate after 4M steps). This advantage likely stems from IQR's robustness to outliers in Q-value estimates during early training when ensemble variance is high. According to our experimental results, the effect of $\alpha$ depends on task and demonstration quality, but the performance of our SPReD-E is not very sensitive to the choice of $\alpha$. There is no general trend indicating that a smaller or larger value of $\alpha$ would be better. This is a task-specific hyperparameter which can be tuned to achieve better performance for a specific task. In our work, we take $\alpha = 10$ for the consistent comparisons, which performs well overall.

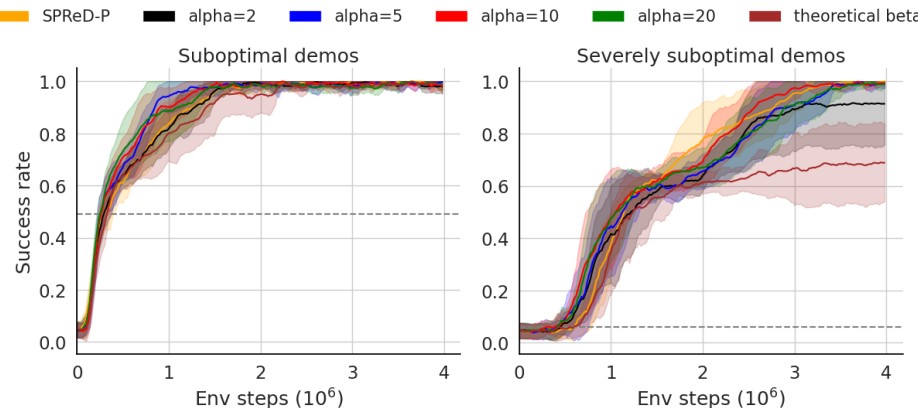

Figure 12: Effect of normalisation constant $\alpha$ of SPReD-E in the FetchPickAndPlace environment with suboptimal and severely suboptimal demonstrations (success rates shown as dashed lines). The performances of SPReD-P SPReD-E with theoretical $\beta = \sigma\sqrt{2\pi}$ are given as the baseline of comparison.

## H  Broader impacts

Our research on uncertainty-aware reinforcement learning from demonstrations offers several societal benefits: accelerating robotic automation across industries, enabling safer operation in hazardous environments, and democratizing access to robotic solutions through reduced demonstration requirements. However, these advances may also contribute to workforce displacement in sectors reliant on manual labor, potentially devalue certain specialised demonstration skills, and introduce challenges for accountability in systems that require minimal human supervision. We acknowledge that complementary workforce development programs and economic policies will be important alongside technological advances in automation to address potential negative impacts on employment.

