# OpenReview forum: "Uncertainty-Based Smooth Policy Regularisation for Reinforcement Learning with Few Demonstrations"
_NeurIPS.cc/2025/Conference — NeurIPS 2025 poster_

### Official Review · Reviewer_hmL7 · 2025-06-12

**Clarity:** 3
**Significance:** 3
**Originality:** 3
**Rating:** 5
**Confidence:** 4

**Summary:**

This paper tackles the setting of demonstration-guided RL and when the agent should imitate a demonstration versus prioritizing the RL objective based on the estimated Q-values.  The authors propose an uncertainty-aware, smooth regularization weight which has lower gradient variance compared to prior methods that apply a binary weight.  They learn an ensemble of Q-functions and propose two variants of their method: 1) based on the probability that the demonstration Q-value is greater than the policy's and 2) based on the magnitude of difference between the two Q-values.  The theoretical results demonstrate that continuous regularization weights result in lower gradient variance and characterize the behavior of the weights.  The empirical results show their method outperforms other binary weighting methods in multiple manipulation tasks.

**Questions:**

1. Why did you not include experiments in other domains or use some of the benchmarks from prior work?

2. Why did you use different quality demonstrations for each task?  Was the performance sensitive to how sub-optimal the demonstration were depending on task?

3. Why did you pick 1 million timesteps for the results in Table 1?  Given the learning curves in Figure 1, Table 1 does not seem necessary, especially since each task seems to converge at a different rate.

**Ethical Concerns:**

["NO or VERY MINOR ethics concerns only"]

**Final Justification:**

I raised my score from a 4 to 5 since my main concern was that the experiments were all done in one domain.  The authors provided additional experiments in locomotion tasks in the rebuttal which addressed this concern.

**Limitations:**

Yes

**Paper Formatting Concerns:**

No formatting issues

**Quality:**

2

**Strengths And Weaknesses:**

Strengths:
1. Identified a clear limitation of prior methods in doing binary regularization weighting for demonstrations.
2. Methods are sound for the most part and theoretical results provide some understanding for how they work.
3. Strong empirical results.
4. Well-presented and written.

Weaknesses:
1. The derivation and contribution of SPReD-E are unclear to me.  The paper claims that this measure should "respect the distributional nature of our estimates" and "compare $x$ in the support of $\mu$ to $y$ in the support of $v$", suggesting that this measure should be defined probabilistically.  However, the final measure is a simple difference between means, so it cannot really be called uncertainty-aware.

2. My main concern is that this paper lacks evaluation in other domains.  All the results are on OpenAI manipulation tasks.   Meanwhile the prior works the authors cited had experiments in D4RL, locomotion, and game tasks.  This is the main reason I am on the fence for acceptance.

3. It's unclear to me what the contribution of the ensemble is versus the continuous regularization weights, especially for SPReD-E.  Do you have results ablating the two?

---

> ### Author Rebuttal · Authors · 2025-07-30
>
> We sincerely thank you for your thoughtful review and constructive feedback. We appreciate your recognition of our key contribution—identifying and addressing the limitation of binary regularisation in demonstration-based RL—as well as your positive assessment of our theoretical analysis and empirical results. Your acknowledgment that our methods are sound and the paper is well-presented is particularly encouraging. We address your concerns and questions below to further strengthen our contribution.
>
> **SPReD-E's uncertainty-aware nature:** We appreciate the reviewer's careful reading of our derivation. While the final formulation $A = \mathbb{E}[Q_d] - \mathbb{E}[Q_π]$ is indeed a difference of means, this emerges from a principled approach that considers the distributional nature of our estimates. In the end, we need a partial ordering on the set of probability measures. We imagine there are many such orderings that are more exciting than the one we have used here. However, we justify our choice based on (1) our belief it is better to start simpler and (2) empirical performance of the method suggests it is state-of-the-art.
>
> SPReD-E incorporates uncertainty through the normalisation term $β$, which captures the spread of both Q-value distributions:
>
> **$β = α × \frac{1}{2}[IQR(Q(s_d,a_d)) + IQR(Q(s_d,π(s_d)))]$.**
>
> This makes SPReD-E fundamentally uncertainty-aware:
>
> **$p_E = clip(e^{A/β} - 1, 0, 1)$**
>
> The key insight is that the same advantage $A$ produces different weights depending on our estimation uncertainty:
>
> - **High uncertainty** (large $β$): $e^{A/β} ≈ 1 + A/β$ → small weights (conservative)
> - **Low uncertainty** (small $β$): $e^{A/β}$ grows rapidly → larger weights (confident)
>
> **Example**: For $A = 2.0$:
>
> - High uncertainty ($β = 10$): $p_E = e^{0.2} - 1 ≈ 0.22$
> - Low uncertainty ($β = 1$): $pE = e^{2} - 1 ≈ 6.39 → 1.0$ (after clipping)
>
> The transport map formulation (lines 206-211) provides the theoretical foundation for why comparing distributions reduces to comparing means when using optimal transport. While the final implementation is computationally simple, this derivation justifies why $A = \mathbb{E}[Q_d] - \mathbb{E}[Q_π]$ is a principled choice for measuring distributional differences. The uncertainty-awareness comes from scaling this advantage by the combined uncertainty of both distributions.
>
> **Limited domain evaluation:**  We appreciate your interest in seeing broader domain evaluation. Our choice of manipulation tasks was deliberate and well-motivated—these environments feature sparse rewards and precise control requirements where demonstration guidance provides the most value, making them ideal for evaluating our method. Our evaluation includes all robotic manipulation tasks tested in the most relevant baseline—Q-filter and hand manipulation tasks in addition.
>
> D4RL, used by methods like RLPD, serves a fundamentally different purpose. It provides massive offline datasets (1-2M transitions) for pure offline learning, whereas SPReD tackles online RL with a relatively small number of demonstrations (~5K transitions). Testing SPReD on D4RL would require either:
>
> - Discarding 99.5% of the data to match our sparse demonstration setting
> - Abandoning our core research question of learning from limited demonstrations
>
> Neither approach would meaningfully evaluate our contribution. Notably, our main baseline Q-filter was also evaluated exclusively on manipulation tasks, providing appropriate comparison.
>
> To address your comments, we have now expanded our evaluation to Hopper, HalfCheetah and Walker2d in locomotion domain to demonstrate generalisation:
>
> | Task | TD3 | EnsTD3 | Q-filter | EnsQ-filter | RLPD | SPReD-P | SPReD-E |
> | --- | --- | --- | --- | --- | --- | --- | --- |
> | Hopper | 1468 ± 849 | 2144 ± 966 | 2818 ± 438 | 2930 ± 546 | 2744 ± 580 | **3246 ± 18** | 2740 ± 344 |
> | HalfCheetah  | 3728 ± 419 | 3775 ± 1756 | 4671 ± 760 | 7188 ± 934 | 6721 ± 3542 | **8060 ± 171** | 7336 ± 601 |
> | Walker2d  | 1861 ± 1159 | 2334 ± 772 | 1485 ± 589 | 3576 ± 528 | **4519 ± 154** | 3351 ± 842 | 2403 ± 1154 |
>
> The table presents scores at 200K steps to assess the sample efficiency. Though EnsQ-filter without continuous regularisation and RLPD achieve competitive performance in these environments with dense rewards and near-expert demonstrations, SPReD achieves high performance for all three tasks and is particularly outstanding on HalfCheetah (SPReD-P gains 12% improvement from EnsQ-filter and 72% improvement from Q-filter), confirming that our uncertainty-aware approach effectively transfers across different continuous control domains.
>
> While we believe our current evaluation comprehensively validates our approach, we agree that exploring additional domains is helpful and will include more results (e.g., other locomotion tasks Ant and Humanoid with demonstrations) with learning curves in our final manuscript.
>
> **Contribution of ensemble vs. continuous regularisation:** We totally agree—since SPReD introduces both continuous weights and ensemble-based uncertainty estimation, it's important to understand their individual and combined effects. We conducted a systematic ablation study to isolate these contributions. By varying ensemble size (2 vs. 10) and regularisation type (binary vs. continuous), we can assess each component's impact (with success rates at 1M steps):
>
> |  | Q-filter | EnsQ-filter | SPReD-E (Ensemble size  2) | SPReD-P (Ensemble size  2) | SPReD-E (Ensemble size 10) | SPReD-P (Ensemble size  10) |
> | --- | --- | --- | --- | --- | --- | --- |
> | FetchPickandPlace | 0.584 ± 0.041 | 0.648 ± 0.039 | 0.120 ± 0.240 | 0.712 ± 0.064 | 0.840 ± 0.119 | 0.744 ± 0.103 |
>
> Our key findings are as follows:
>
> - *Continuous regularisation is the primary driver:* Even with a minimal ensemble (size 2), SPReD-P improves over Q-filter by 22% (0.712 vs 0.584). This demonstrates that smooth, uncertainty-proportional weights are fundamentally better than binary decisions, regardless of ensemble size.
> - *Ensembles add complementary value*: Expanding from 2 to 10 critics further boosts performance for both SPReD variants. However, the gains are method-specific—SPReD-P shows modest improvement (+4%, 0.712 vs 0.744) while SPReD-E shows dramatic improvement (+24%, 0.120 vs 0.840 ).
> - *The methods have different uncertainty requirements:* This differential benefit reveals an important insight: SPReD-E's exponential weighting critically depends on accurate uncertainty estimates through $β$. With only 2 critics, the IQR calculation is noisy, leading to suboptimal scaling. In contrast, SPReD-P's probabilistic approach is inherently more robust to limited ensemble sizes.
>
> The ablation confirms that while both components contribute independently, they work synergistically. Continuous regularisation provides the foundation for better learning, while larger ensembles enable more precise uncertainty quantification—particularly crucial for SPReD-E's exponential scaling mechanism. This validates our design decision to combine both innovations rather than pursuing either in isolation. We will include this ablation study, with full learning curves, in the revised manuscript to make these contributions explicit.
>
> ### Questions
>
> **Q1:**  Please see our response to Weakness 2 above, where we present new locomotion results and explain why benchmarks like D4RL (designed for offline RL with 1-2M transitions) are incompatible with our sparse demonstration setting.
>
> **Q2:**  We deliberately varied demonstration quality across tasks to evaluate robustness in realistic scenarios where expert demonstrations may not always be available.
>
> To directly address sensitivity: Figure 2 systematically compares all methods across three quality levels (expert, suboptimal, severely suboptimal) on fixed tasks. SPReD consistently outperforms baselines at every quality level, demonstrating robustness rather than sensitivity. Even with 99% random demonstrations mixed with 1% expert data, SPReD methods successfully filter out noise and achieve reasonable performance, while baselines fail catastrophically.
>
> This robustness is by design—our uncertainty-aware weighting naturally adapts to demonstration quality. Poor demonstrations lead to low advantage and high uncertainty with respect to Q-estimates, resulting in smaller weights that diminish their influence. All demonstration-based methods perform better with higher-quality data, but SPReD consistently outperforms other demonstration-based methods and is not misled by low-quality demonstrations.
>
> **Q3:**  Table 1 provides a standardised comparison point, following prior fundamental work (e.g., Table 1 in TD3 [1], Table S3 in A3C[2]). This enables fair comparison across methods and reproducible benchmarking for future work. We chose 1M steps as it allows meaningful learning progress while remaining computationally tractable for all baselines.
>
> The combination of Table 1 and Figure 1 provides complementary information: Table 1 enables quick numerical comparison of sample efficiency, while Figure 1 shows complete learning dynamics and convergence behaviour. Together, they offer a more comprehensive view than either would individually.
>
> ### References
>
> [1] Fujimoto, S., Hoof, H., Meger, D. Addressing function approximation error in actor-critic methods. In *International conference on machine learning*, pages 1587–1596. PMLR, 2018.
>
> [2] Mnih, V., Badia, A.P., Mirza, M., et al.  Asynchronous methods for deep reinforcement learning. In *International conference on machine learning*, pages 1928-1937. PMLR, 2016.

---

> ### Comment · Reviewer_hmL7 · 2025-08-01
>
> Thank you for the thorough response, especially the clarification on SPReD-E's derivation and the new ablation on ensemble size, which addressed my concerns on those topics.
>
> On the limited evaluation domain: Thanks for providing an explanation for the tasks chosen in the paper and adding the new experiments.  This did bring up some additional questions for me.  You claim that you chose manipulation tasks because they have sparse rewards and require precise control.  I believe you can create similar settings in other domains like navigation/goal reaching for locomotion agents, or even manipulation tasks with a different robot or simulator.  Furthermore, utilizing a small subset of D4RL as demonstrations, as the authors suggest, would be a reasonable way of evaluating their method.  So I do not see a reason to evaluate this method in only one manipulation domain.  However, given the new Hopper, HalfCheetah, and Walker2D results, I will increase my score to a 5.

---

> > ### Author Response · Authors · 2025-08-03
> >
> > Thank you for your positive feedback and thoughtful comments. We sincerely appreciate your recognition of our clarifications and additional experiments.

---

### Official Review · Reviewer_npaw · 2025-06-30

**Clarity:** 4
**Significance:** 3
**Originality:** 3
**Rating:** 5
**Confidence:** 3

**Summary:**

The paper proposes a novel method for combining demonstrations with online off-policy reinforcement learning. This is motivated by the observation that existing methods, like Q-filter, make hard, binary decisions about when to imitate demonstrations based on point estimates of Q-values. The authors instead propose to learn an ensemble of Q-functions to capture epistemic uncertainty. Deciding whether to imitate a demonstrated action can then be phrased as comparing the distributions of Q-values of the demonstrated and policy action induced by the Q-function ensemble. Two variants of this idea are presented: SPReD-P assumes Q-values follow a Gaussian distribution and computes behavioral cloning (BC) weights from the probablity that the mean of demonstration Q-values is higher than the mean of the policy Q-values. SPReD-E computes the BC weights based on an exponential of the difference of the means of the two Q-value distributions, normalized by their interquartile range. Both variants result in continuous BC weights that take the uncertainty of the Q-value estimates into account. A theoretical analysis shows that this reduces the variance of the BC weights compared to Q-filter, and that the weights behave intuitively in different limits. An experimental evaluation demonstrates that both versions of SPReD outperforms competitive baselines on a range of continuous control tasks.

**Questions:**

### Questions
* Why is there no initial state distribution mentioned in the “Setup” paragraph (line 108)? In line 114 the state is sampled from $E$ but $E$ is never introduced, I think.
* In line 261 f. in Property 5.3, the statement “both weighting schemes implement similar degrees of caution” seems a bit odd because at $A=0$ on of them has weights 0.5 and the other one 0.0.
### Suggestions
* The sentence “Adapting to continuous spaces, policy with soft constraint from demonstrations [11] introduces substantial computational complexity.” in line 31 sounds awkward.
* The learning curves look a bit crowded. As a result, it’s sometimes hard to discern which method is better. Smoothing them before plotting (and noting so somewhere) could help.
* Line 112: “in replay buffer $R$", forgot article. Also $R$ is already used for the reward function so it might be better to choose another symbol.
* Also in line 112: “with exploration noise added to actions”. This sounds like the noise is only added when saving the actions to the replay buffer. However, this is not correct, right?
* In Property 5.1 in line 241 an $\alpha$ makes an appearance. Should this be a $\beta$?
* The caption of Figure 1 states that the SPReD methods are shown in red and purple but it’s actually red and brown as far as I can see.

**Ethical Concerns:**

["NO or VERY MINOR ethics concerns only"]

**Final Justification:**

I recommend acceptance of the submission for the following reasons:
* The contributions of the paper in the area of combining learning from demonstrations and online interactions are significant, well-motivated, and relevant to the community.
* The paper is well-written.
* The empirical results are strong.
* The theoretical analysis matches the proposed algorithm variations and help understand their properties.

During the rebuttal the authors adressed my questions and outlined revisions to the camera-ready paper that will improve consistency and presentation of the paper. All points raised in my review were sufficiently addressed.

**Limitations:**

Yes, the authors discuss limitations in the conclusion.

**Quality:**

3

**Strengths And Weaknesses:**

### Strengths:
* The paper is well-written and structured.
* The motivation for the proposed algorithm is made clear early on.
* The theoretical analysis is a good addition to the paper.
* The empirical results are strong, in particular on challenging tasks like stacking, and back the claims in the introduction.
* The ablation studies help understand how SPReD and the baselines perform for different demonstration qualities and sample sizes.
### Weaknesses:
* The proposed method builds on Q-filter but uses smooth, uncertainty-aware BC weights. That makes it somewhat incremental. However, the clear motivation for doing this and the strong empirical results make the paper relevant nevertheless.

---

> ### Author Rebuttal · Authors · 2025-07-30
>
> We sincerely thank you for your thorough review and positive assessment of our work. We appreciate your recognition of our contributions and your constructive feedback that helps improve the clarity of our presentation. Below, we address your specific questions and incorporate your helpful suggestions.
>
> ### Questions
>
> **Q1: Missing initial state distribution (line 108).** Thank you for identifying this oversight. We have added the initial state distribution $ρ_0$ to the MDP framework in the Setup paragraph. To clarify the notation in line 114: $E$ represents the environment defined by our MDP, and $s_0 \sim ρ_0$ denotes sampling the initial state from this distribution. The complete MDP is now properly defined as $\mathcal{M} = (\mathcal{S}, \mathcal{A}, P, R, γ, ρ_0)$, ensuring notational consistency throughout the paper.
>
> **Q2: Property 5.3 - apparent inconsistency at $A=0$.**  We appreciate you highlighting this potentially confusing point. While it may seem odd that $p_P = 0.5$ and $p_E = 0$ at $A=0$, this reflects their different interpretations of uncertainty:
>
> - **SPReD-P** (probabilistic): When $A=0$, the Gaussian distributions overlap completely, yielding $\mathbb{P}(Q_d > Q_π) = 0.5$.
> - **SPReD-E** (exponential): Using $e^{A/β} - 1$, which equals 0 when $A=0$, enforces imitation only for positive advantages.
>
> The key insight from Property 5.3 is that despite this difference at $A=0$, both methods exhibit similar *rates of change* near $A=0$. Specifically, when $β = σ\sqrt{2π}$, we show that $p_P - 1/2 ≈A/β≈ p_E$, meaning both follow similar trends when advantages are small compared to the uncertainty. This relationship is visualised in Figure 7 (Appendix, page 22). We have clarified this interpretation in the main text.
>
> ### Suggestions
>
> - **Line 31**: Revised to “Jing et al. [11] treat demonstrations as a soft constraint on policy exploration, formulating a constrained policy optimisation problem. Although they reduce overhead by applying a local linear search on its dual, the approach still involves considerable computational complexity.”
> - **Learning curves**: We have applied 5-point moving average smoothing to improve readability and noted this preprocessing in all figure captions.
> - **Notation**: Changed replay buffer notation from $\mathcal{R}$ to $\mathcal{B}$ throughout to avoid conflict with the reward function.
> - **Exploration noise**: Clarified that noise is added in two contexts: (1) during action execution for exploration, and (2) during critic target computation for smoothing (both following TD3 design).
> - **Property 5.1**: Corrected to use $α$, which is the normalisation constant in $β = α ×$ IQR as defined in Appendix C (page 18). We have improved consistency between the methodology section and theoretical properties.
> - **Figure 1 caption**: Corrected to accurately describe SPReD methods as "red and brown" lines.
>
> All the corrections and clarifications mentioned above have been incorporated into our revised manuscript. Since we cannot upload a revised PDF during the rebuttal phase, these changes will be reflected in the camera-ready version if the paper is accepted. We are happy to provide any additional clarifications if needed.

---

> > ### Comment · Reviewer_npaw · 2025-08-07
> >
> > Thank you for your detailed response to my questions and concerns. Please accept my apologies for my late response.
> >
> > I appreciate the summarized revisions that will be incorporated in the camera-ready version and the clarifications regarding my question about the weights of the two variants of the method. My concerns and questions have been addressed completely.

---

> > > ### Author Response · Authors · 2025-08-07
> > >
> > > Thank you for your kind message. We are glad to hear that your concerns have been addressed.

---

### Official Review · Reviewer_JFdB · 2025-07-05

**Clarity:** 3
**Significance:** 2
**Originality:** 2
**Rating:** 4
**Confidence:** 3

**Summary:**

The paper introduces Smooth Policy Regularisation from Demonstrations (SPReD), an ensemble-based extension of the Q-filter that turns the binary “imitate / ignore” rule into a continuous weight proportional to the critic ensemble’s epistemic uncertainty. Two instantiations are proposed:

- SPReD-P: probabilistic weighting via the likelihood that a demo action’s Q-value exceeds the policy action’s.

- SPReD-E: exponential weighting that scales with an advantage-to-uncertainty ratio.

The authors prove lower gradient variance than the Q-filter and present large empirical gains on eight MuJoCo manipulation tasks with sparse rewards.

**Questions:**

Besides the points of Weaknesses:

- How does SPReD-E perform if β is set using the theoretical scaling (β = σ√2π) instead of a heuristic IQR-based α?
- Is it possible to use dropout-based uncertainty or bootstrapping to reduce compute overhead, and how does performance compare?
- Why was AWAC (with HER) not included as a baseline? Please report AWAC on at least FetchStack2.

**Ethical Concerns:**

["NO or VERY MINOR ethics concerns only"]

**Final Justification:**

The rebuttal results have addressed most of my concerns. I recommend authors to include the results in the revision. And I am happy to increase my score.

**Limitations:**

The paper included limitations in the conclusion section.

**Paper Formatting Concerns:**

No.

**Quality:**

2

**Strengths And Weaknesses:**

**Strengths**
- The need for smoother, uncertainty-aware demonstration regularization is well-motivated. Lower variance in policy updates is shown both theoretically and empirically.
- Only requires minor changes to TD3+HER setups; ensemble critics are used both for targets and uncertainty estimates.
- Results are robust across eight tasks, with substantial improvements, especially on harder manipulation and stacking environments.

**Weaknesses**
- The paper does not cite key related methods: IQL [1], UWAC [2], which combine advantage-weighted or uncertainty-weighted imitation in RL.
- The cost of ensembles (10 critics) is not directly compared against single-critic baselines. Throughput is mentioned, but energy and wall-time under fixed hardware are not reported.
- The approach is only tested in MuJoCo. No robustness analysis under sim-to-real gap, nor any explicit discussion of failure cases under noisy or misaligned reward signals.

**References**
[1] "Offline Reinforcement Learning with Implicit Q-Learning". ICLR 2022.
[2] "Uncertainty Weighted Actor-Critic for Offline Reinforcement Learning". ICML 2021.

---

> ### Author Rebuttal · Authors · 2025-07-30
>
> Thank you for your constructive feedback. We address each of your concerns below:
>
> **Missing citations:** Thank you for highlighting these methods. We will include IQL and UWAC in our related work section. While both share ideas with SPReD, such as advantage weighting (IQL) and uncertainty-based weighting via dropout (UWAC), they address different problems:
>
> - **IQL** focuses on offline and offline-to-online RL, using expectile regression to learn a value function and performing advantage-weighted policy extraction to learn a policy without explicit bootstrapping from the policy.
> - **UWAC** is an offline RL method that weights actor updates using dropout-based uncertainty to manage out-of-distribution actions.
>
> In contrast, **SPReD** tackles online RL with few demonstrations (~5000 transitions), where the agent must continuously balance exploration and selective demonstration guidance. Its core contribution is determining when demonstrations remain useful during ongoing learning.
>
> We will include them in our revised related work section with a clear discussion of the differing problem settings.
>
> **Computational cost analysis:** We have included computational cost analysis in Appendix C (Figure 4, page 18). Our experiments on a single GeForce GTX 3090 GPU show:
>
> - **Wall-time**: SPReD requires ~2.6 hours per 4M environment steps, nearly identical to TD3 (2 critics), while RLPD (also 10 critics) requires ~4.8 hours.
> - **Throughput**: SPReD processes ~427 environment steps/second, compared to TD3's ~444 steps/second—only a 3.8% decrease despite using $5×$ more critics.
> - **Energy**: While we did not measure energy consumption directly (as is standard in RL papers), the similar wall-time to TD3 suggests comparable energy usage.
>
> The key to our efficiency is *vectorised* critic implementation: instead of processing 10 critics sequentially, we stack their computations into batched tensor operations that execute in parallel on GPU. This is why SPReD achieves nearly the same throughput as TD3 despite using $5×$ more critics. Given that SPReD achieves up to $14×$ better success rates than baselines on complex tasks (Table 1), this $<4$% throughput cost is well justified. We will add these throughput metrics to the main paper for transparency.
>
> **Regarding sim-to-real gaps and reward misspecification:** While important, they remain orthogonal to our contribution. In most online RL research, the rewards are assumed to be accessible and reliable, and we continue with this assumption since misleading reward is not the primary problem we want to address in this work. If you deem the consideration of it necessary, we are willing to do the experiments with noisy rewards to check the robustness and add the results to the final manuscript. Our uncertainty-aware approach should actually provide more robustness than binary methods under such conditions. We will update our paper to clarify sim-to-real gaps as remaining limitations in the appropriate section and regard this as future work.
>
> ### Questions
>
> **Q1**: We evaluated both approaches on multiple tasks and gave the success rate at 1M steps in the table. The results show that our IQR-based $β$ (with $α=10$) consistently outperforms the theoretical scaling:
>
> | Task | SPReD-E ($α=10$) | SPReD-E ($β=σ\sqrt{2π}$) |
> | --- | --- | --- |
> | FetchPush | 0.984 ± 0.032 | 0.976 ± 0.020 |
> | FetchPickAndPlace (Suboptimal demos) | 0.840 ± 0.119 | 0.720 ± 0.104 |
> | FetchPickAndPlace (Severely suboptimal demos) | 0.304 ± 0.194 | 0.088 ± 0.059 |
>
> The theoretical scaling $β=σ\sqrt{2π}$ builds a connection with our probabilistic advantage weighting method SPReD-P, but there is no guarantee that it works the best. While both $σ$ and IQR measure Q-value distribution spread, the IQR-based approach provides better empirical performance, particularly in more challenging tasks (17% improvement in FetchPickAndPlace). The advantage is even more pronounced with lower-quality demonstrations (44% higher success rate after 4M steps). This advantage likely stems from IQR's robustness to outliers in Q-value estimates during early training when ensemble variance is high. Additionally, the hyperparameter $α$ allows task-specific tuning as shown for FetchPickAndPlace in Appendix G (Figure 9, page 24), whereas the theoretical scaling is fixed. We will include complete learning curves comparing both approaches in the revised manuscript.
>
> **Q2:** Yes, we investigated dropout-based uncertainty as an alternative to our ensemble approach. Contrary to the intuition that dropout might reduce overhead, our experiments show it actually increases computational cost while degrading performance:
>
> | Method | Time (4M steps) | Success Rate (1M steps) |
> | --- | --- | --- |
> | SPReD-P (Ensemble size10) | 2.6h | 0.744 ± 0.103 |
> | SPReD-P (Dropout) | 20.3h | 0.600 ± 0.044 |
>
> *Dropout configuration: rate=0.1, 500 forward passes per critic (2 critics).*
>
> The **dropout approach is 8× slower** due to requiring 1000 stochastic forward passes (500 × 2) to estimate uncertainty, while our ensemble uses a single vectorised pass through 10 critics in parallel on GPU. Although both methods eventually learn an expert policy within 4M steps, dropout shows worse sample efficiency (19% drop), likely due to: (1) noisier uncertainty estimates, (2) overconfident predictions on unseen data [1], and (3) computational bottlenecks from repeated passes that hinder efficient batch processing.
>
> These results are consistent with prior findings [1] that ensembles yield better uncertainty estimates than dropout, and modern GPUs enable highly efficient ensemble parallelisation.
>
> Bootstrapping presents other computational challenges: (1) separate data samples per model prevent batch-sharing, (2) models process different minibatches, blocking parallelisation, and (3) storing multiple bootstrap samples increases memory demands. In contrast, our ensemble processes the same batch across all critics via a single vectorised operation, achieving near-linear speedup. Moreover, Bootstrapped DQN [2] supports this strategy, suggesting that diversity from random initialisations of deep NN eliminates the need of explicit data bootstrapping.
>
> These results confirm that vectorised ensembles offer the best balance between uncertainty estimation quality and computational efficiency.
>
> **Q3:** We have added AWAC+HER as a baseline (with and without pretraining). We first compare our method and baselines with AWAC on locomotion tasks where AWAC has good performance [3]. The demonstrations contain 5 suboptimal trajectories ($≤ 5000$ transitions) for each task. The table shows scores at 0.2M steps for locomotion tasks, success rates at 10M steps for FetchStack2 and 1M steps for other fetch tasks.
>
> | Task | AWAC | AWAC  (pretraining) | AWAC (resample demos) | SPReD-P | SPReD-E |
> | --- | --- | --- | --- | --- | --- |
> | Hopper | 2855 ± 707 | **3176 ± 94** | 2461 ± 1041 | **3246 ± 18** | 2740 ± 344 |
> | HalfCheetah  | 4835 ± 854 | 4310 ± 963 | 5115 ± 823 | **8060 ± 171** | 7336 ± 601 |
> | Walker2d  | **3987 ± 108** | **3980 ± 353** | **4005 ± 433** | 3351 ± 842 | 2403 ± 1154 |
> | FetchPush | 0.056 ± 0.020 | 0.056 ± 0.032 | 0.072 ± 0.030 | **0.976 ± 0.020** | **0.984 ± 0.032** |
> | FetchSlide | 0.000 ± 0.000 | 0.000 ± 0.000 | 0.016 ± 0.020 | 0.112 ± 0.096 | **0.240 ± 0.044** |
> | FetchPickandPlace | 0.048 ± 0.030 | 0.040 ± 0.025 | 0.120 ± 0.080 | **0.744 ± 0.103** | **0.840 ± 0.119** |
> | FetchStack2 | 0.008 ± 0.016 | 0.008 ± 0.016 | 0.008 ± 0.016 | 0.840 ± 0.110  | **0.920 ± 0.057** |
>
> SPReD method surpasses all variants of AWAC for HalfCheetah. AWAC is competitive for Hopper and Walker2d with near-expert demonstrations. However, the learning score of AWAC is asymptotically lower than SPReD for all three locomotion tasks we tested, which will be visualised with the figure of learning curves in our final manuscript.
>
> Although AWAC achieves moderate performance in locomotion tasks, our experiments show that AWAC struggles significantly in our setting with few demonstrations and environments with sparse rewards. To give AWAC the best chance, we also implemented a variant that resamples demonstrations during online training (similar to our approach).  However, the benefit of continuously sampling demonstrations is negligible for AWAC on both locomotion and manipulation domains. SPReD's uncertainty-aware weighting significantly outperforms AWAC because it explicitly reasons about when demonstrations remain useful rather than relying on advantage estimates from limited data. AWAC's poor performance stems from a fundamental mismatch with our problem setting:
>
> 1. **AWAC assumes large offline datasets** (~1M transitions) for effective pretraining, but we have only ~5K demonstration transitions. Without sufficient pretraining data, and since demonstrations are quickly diluted in the replay buffer, their impact fades early, effectively reducing AWAC to standard RL.
> 2. **AWAC's advantage weighting** assumes the offline data covers a substantial portion of the state space, which doesn't hold with sparse demonstrations
>
> We will include complete AWAC results on other tasks and learning curves in the revised manuscript.
>
> ### References
>
> [1] Lakshminarayanan, B., Pritzel, A., Blundell, C. Simple and scalable predictive uncertainty estimation using deep ensembles. *Advances in Neural Information Processing Systems*, 2017, 30.
>
> [2] Osband, I., Blundell, C., Pritzel, A., et al. Deep exploration via bootstrapped DQN. *Advances in neural information processing systems*, 2016, 29.
>
> [3] Nair, A., Gupta, A., Dalal, M., et al. Awac: Accelerating online reinforcement learning with offline datasets. *arXiv preprint arXiv:2006.09359*, 2020.

---

> > ### Comment · Reviewer_JFdB · 2025-08-04
> >
> > I appreciate authors for providing detailed experiments. They have addressed most of my concerns. I recommend authors to include the results in the revision. And I am happy to increase my score.

---

> > > ### Author Response · Authors · 2025-08-07
> > >
> > > Thank you for your positive feedback  and for considering an increase in your score. We will include the results in the revised version.

---

### Note · Authors · 2025-08-15

We thank the reviewers and the area chair for your thoughtful consideration and feedback on our work. We are pleased that our rebuttal addressed the majority of concerns and provided improved clarifications accordingly. Our additional results continue to demonstrate the strength of our approach across the evaluated tasks, including Humanoid and Ant, which we completed after the rebuttal. All these results are included in our revised manuscript. We believe the paper is now well-motivated, well-structured and clearly presented, supported by strong empirical and theoretical results.

---

### Decision · Program_Chairs · 2025-09-17

**Decision:**

Accept (poster)

**Comment:**

Summary: The paper introduces SPReD (Smooth Policy Regularisation from Demonstrations), a framework for reinforcement learning with demonstrations that replaces binary Q-filter decisions with continuous, uncertainty-aware weighting. Two variants are proposed: SPReD-P, which uses probabilistic estimates of demonstration superiority, and SPReD-E, which applies exponential advantage weighting. Theoretical analysis shows reduced gradient variance and adaptive down-weighting of suboptimal demonstrations, while empirical results across eight robotics tasks demonstrate strong improvements over TD3, Q-filter, EnsQ-filter, and RLPD, with gains especially pronounced in complex stacking environments.

We received three expert reviews, with scores, 4, 5, 5, and the average review is 4.67. The reviewers appreciated theoretically grounded approach, including the variance reduction guarantees and properties showing adaptive handling of uncertainty and suboptimal demonstrations. The reviewers also appreciated strong empirical results. Additional experiments provided during rebuttal (locomotion benchmarks) strengthened generality claims. The paper is well-written, logically structured, and supported by ablations on demonstration quality and size.  While the reviewers raised some concerns and questions, the authors have successfully addressed all of them, and the reviewers have acknowledged and appreciated this effort.

Overall, the paper offers a principled and effective contribution to RL with demonstrations. While not a significant departure from prior work, the demonstrated improvements and theoretical grounding make it a valuable contribution for the community.